# Global-aware Beam Search for Neural Abstractive Summarization

**Ye Ma**[1,4]     **Zixun Lan**[2,4]     **Lu Zong**[1,4*]     **Kaizhu Huang**[3]

[1] Department of Financial and Actuarial Mathematics, School of Science
[2] Department of Applied Mathematics, School of Science
[3] Department of Intelligent Science, School of Advanced Technology
[4] Laboratory for Intelligent Computing and Financial Technology
Xi'an Jiaotong-Liverpool University, SIP, 215123 Suzhou, China

{ye.ma, lu.zong, kaizhu.huang}@xjtlu.edu.cn, zixun.lan19@student.xjtlu.edu.cn

## Abstract

This study develops a calibrated beam-based algorithm with awareness of the global attention distribution for neural abstractive summarization, aiming to improve the local optimality problem of the original beam search in a rigorous way. Specifically, a novel global protocol is proposed based on the attention distribution to stipulate how a global optimal hypothesis should attend to the source. A global scoring mechanism is then developed to regulate beam search to generate summaries in a near-global optimal fashion. This novel design enjoys a distinctive property, i.e., the global attention distribution could be predicted before inference, enabling step-wise improvements on the beam search through the global scoring mechanism. Extensive experiments on nine datasets show that the global (attention)-aware inference significantly improves state-of-the-art summarization models even using empirical hyper-parameters. The algorithm is also proven robust as it remains to generate meaningful texts with corrupted attention distributions. The codes and a comprehensive set of examples are available.[2]

## 1   Introduction

As the barriers exist from the auto-regressive design of neural probabilistic text generators to predicting the global optimum directly [30], the heuristic algorithm beam search that factorizes global optimization to multiple local optimizations, has been popularly used for text decoding [24]. In the original beam search setting, the global optimum is a hypothesis $\mathbf{g}$ of the highest probability among all possible sentences, and consists of words in vocabulary $\mathcal{V}$. Given the global optimum at step $t$ denoted as $\mathbf{g}_{\leq t}$, the local optimums $\mathbf{l}_{\leq t}$ refer to $\mathcal{K}$ candidate sequences with the highest probabilities at each step. While it is necessary to compromise on the beam size $\mathcal{K} \ll \mathcal{V}$ to ensure text quality [3, 25, 24] and search efficiency, beam search suffers from a major limitation due to its local property. Concretely, assuming that the global optimal hypothesis is within the $\mathcal{K}$ local optimal hypotheses of the highest probabilities, i.e. $p(\mathbf{g}_{\leq t}) \geq p(\mathbf{l}_{\leq t})$, for all $t$ until the termination $T$, it operates solely with the local information available at each step. In practice, such assumption may however fail in the case that the probability of the global optimum at step $\tau < T$ is less than those of the local optimums, i.e. $p(\mathbf{g}_{\leq \tau}) < p(\mathbf{l}_{\leq \tau})$, but is adjusted to a higher level in the later steps, $p(\mathbf{g}_{>\tau}|\mathbf{g}_{\leq \tau}) > p(\mathbf{l}_{>\tau}|\mathbf{l}_{\leq \tau})$ and $p(\mathbf{g}_{\leq \tau})p(\mathbf{g}_{>\tau}|\mathbf{g}_{\leq \tau}) > p(\mathbf{l}_{\leq \tau})p(\mathbf{l}_{>\tau}|\mathbf{l}_{\leq \tau})$. This often leads beam search to get stuck in the local optimum from step $\tau$ onward in generating texts.

---

[*]corresponding author
[2]https://github.com/yema2018/global_aware

35th Conference on Neural Information Processing Systems (NeurIPS 2021).

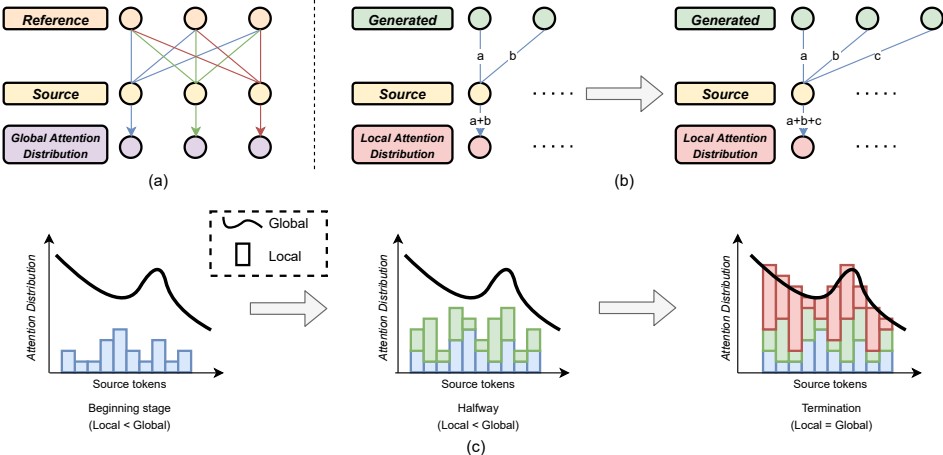

Figure 1: (a) Attention distribution is composed of the summation of cross attention on the same-colored lines, distinguished from that of different-colored lines which always equals 1 due to softmax. (b) Local attention gradually increases as the decoding proceeds. (c) Desired situation: growing local attention has been lower than global attention during decoding and exactly reaches it at the end.

To cope with this limitation, this study proposes a calibrated beam-based algorithm with global awareness at all searching steps. Generally, our novel algorithm is implemented in two phases. Before the beam search (Phase I), the *global attention distribution* is predicted in order to be included as a protocol to calibrate beam search at each step, encouraging the generated hypotheses to attend to the source in a more near-global optimal way. Specifically, the global attention distribution describes how all reference tokens should assign the attention to each source token (illustrated in Figure 1.a), which could be predicted from the source by training an attention-prediction model. The training is fairly straightforward and resembles a sequence tagging task [12], except that the predicted attention distribution from the source is a regression result. There are several advantages of using the attention distribution as the global protocol. 1) Attention distributions are sensitive to the decoder input, suggesting that any input to the decoder leads to a unique attention distribution with fixed model parameters; 2) attention distributions are accessible for almost all text generation tasks thanks to the recent advances in attention models [27, 4, 31]; 3) relying on the source only, the global attention distribution can be predicted before beam search, thus offering a rigorous mechanism to calibrate a global-aware beam search.

During beam search (Phase II), we develop a novel *global scoring mechanism* composed of attention scores and length rewards to guide beam search based on the predicted global attention distribution. As one main theoretical result, we show that the attention score can be considered as the probability that generated texts attend to sources in line with the predicted global attention distribution. Specifically, the generated tokens in each step update the local attention distribution to source tokens dynamically, where the attention values grow monotonically as the generation proceeds (see Figure 1.b). Since the desired situation is that the local distribution reaches exactly the global distribution at the terminal step, we regulate the inference by discouraging local attention from exceeding their corresponding predicted global attention at all steps.

With regards to the core target to investigate the possible paradigm that improves beam search with global awareness during decoding, contributions of this study are summarized as follows:

- We argue that the limitation of beam search roots from its defect in finding the global optimal hypothesis. We improve the algorithm by proposing a global protocol to regulate beam search step-by-step. This paper is the first to predict and deploy the global attention distribution to calibrate the inference in a rigorous way, thus returning a hypothesis that attends to source tokens in a more near-global optimal manner. In contrast, previous works [37, 7, 20, 29, 21] try to use attention distributions to improve beam search, but ignore that the global attention distribution is predictable.

- A novel global scoring mechanism is designed to evaluate the generated sequences at each step based on the desired situation described in Figure 1.c. As theoretically justified, its major component can be elegantly integrated into beam search in the form of a probability so that merely $\mathcal{O}(\mathcal{K})$ of the time complexity is increased in each step (see Section § 3.1 for more details).

- The proposed algorithm with global awareness manifests a robust and plug-and-play property in enhancing beam search for neural abstractive summarization. Without requiring any model or parameter modification, the global-aware inference shows excellent performance in generating meaningful texts, even if the attention distribution is corrupted or not of its own. Further, it is identified that summaries generated by global-aware inference are both higher-quality and different from beam search hypotheses (see *Global-aware* in Table 1). More interestingly, we find that the generation style of a dataset could be transferred by the designated global attention distribution. For instance, summaries of higher abstractness for CNN/DM could be generated by only replacing its global attention distribution with a highly abstractive distribution during inference, as presented in *Global-aware*† of Table 1.

- On the empirical side, we show that the proposed global-aware inference can stably and significantly boost two state-of-the-art summarization models BART [17] and PEGASUS [39] to produce higher quality summaries on nine datasets, even if only the empirical hyper-parameters are used.

Table 1: Use BART [17] fine-tuned in CNN/DM to generate summaries. *Global-aware* uses the attention distribution learned from CNN/DM, while *Global-aware*† takes the attention distribution learned from XSUM.

| | |
|---|---|
| *Beam search* | President Obama says climate change is a public health issue that affects all of us . Obama: "No challenge poses more of a public threat than climate change" Obama: "Millions of people would lose their health insurance" if Affordable Care Act is not upheld . Obama: "I am not anticipating the Supreme Court would make such a bad decision" |
| *Global-aware* | President Obama says climate change is a public health issue that affects all of us . He says the average American can do their part to reduce their own carbon footprint . Obama did not appear particularly concerned about the Supreme Court challenge to the Affordable Care Act . |
| *Global-aware*† | President Barack Obama says climate change is a public health issue . He says the average American can do their part to reduce their carbon footprint . |

## 2 Preliminary

The proposed decoding strategy is applied in BART [17] and PEGASUS [39] to perform summarization. BART is a pre-trained seq-to-seq model whose structure essentially follows a vanilla Transformer encoder-decoder [31]. PEGASUS has a similar structure but is pre-trained on a larger dataset differently. Notably, the fine-tuned parameters of both models are downloaded from *Hugging-Face Models*[3] and are fixed in all subsequent operations, where "fine-tuned" means the pre-trained model has been fine-tuned on a specific dataset.

### 2.1 Beam Search

Since the decoder of the seq-to-seq model is an auto-regressive model, the probability of a target sequence $\boldsymbol{y} = (y_0, \cdots, y_t, \cdots, y_T)$ can be factorized to the probabilities conditional on the source $\boldsymbol{x} = (x_1, \cdots, x_i, \cdots, x_n)$ and $\boldsymbol{y}_{<t} = (y_0, \cdots, y_{t-1})$, i.e.,

$$p(\boldsymbol{y}|\boldsymbol{x}) = \prod_{t=1}^{T} p(y_t|\boldsymbol{x}, \boldsymbol{y}_{<t}), \ T \geq 1 \tag{1}$$

When $T = 0$, $\boldsymbol{y} = (y_0)$ and $p(y_0|\boldsymbol{x}) = 1$ because $y_0$ is a fixed start token. Beam search [8] is a decoding strategy to predict a target sequence by maximizing this factorization. Given a vocabulary set $\mathcal{V}$, at each inference step $t$, beam search selects a candidate beam set $\mathcal{B}_t^{\mathcal{K}} = \{\boldsymbol{b}_t^k\}_{k=1}^{\mathcal{K}}$ (where each beam $\boldsymbol{b}_t^k = (b_0^k, \cdots, b_t^k)$ is a candidate sequence) from an all-possible beam set $\mathcal{B}_t$ of size $\mathcal{K} \times |\mathcal{V}|$, namely,

$$\mathcal{B}_t = \left\{\boldsymbol{b}_{t-1}^k \circ v \mid \boldsymbol{b}_{t-1}^k \in \mathcal{B}_{t-1}^{\mathcal{K}}, \ v \in \mathcal{V}\right\} \tag{2}$$

---

[3]https://huggingface.co/models

$$\mathcal{B}_t^{\mathcal{K}} = \left\{ \boldsymbol{b}_t^k \mid \boldsymbol{b}_t^k = \text{argtopk} \left( \log p(\boldsymbol{b}_t | \boldsymbol{x}) \right), \ \boldsymbol{b}_t \in \mathcal{B}_t \right\}, \ t \geq 1 \tag{3}$$

where argtopk$(\cdot)$ outputs $\mathcal{K}$ beams with the highest conditional probability, and $\circ$ is the concatenation operation. Besides, $\mathcal{B}_0^{\mathcal{K}} = \{b_0^k\}_{k=1}^{\mathcal{K}}$ where $b_0^k$ is the start token. By Eq. 1, $\log p(\boldsymbol{b}_t | \boldsymbol{x})$ is an accumulated value. Its calculation can be simplified as:

$$\log p(\boldsymbol{b}_t | \boldsymbol{x}) = \begin{cases} \log p(\boldsymbol{b}_{t-1}^k | \boldsymbol{x}) + \log p(v | \boldsymbol{x}, \boldsymbol{b}_{t-1}^k), & t \geq 2 \\ \log p(v | \boldsymbol{x}, b_0^k), & t = 1 \end{cases} \tag{4}$$

where the value of $\log p(\boldsymbol{b}_{t-1}^k | \boldsymbol{x})$ is computed from the previous step. Therefore, at each step, we only need calculate the condition probability of each token in the vocabulary set.

A beam is terminated after it generates an ending token, and the beam set of $\mathcal{K}$ terminated beams is defined as $\mathcal{Y}$. The final hypothesis $\boldsymbol{y}^*$ is chosen from $\mathcal{Y}$ based on the beam probability normalized by $length^a$ where $a$ is a hyper-parameter of length [37]:

$$\boldsymbol{y}^* = \underset{\boldsymbol{y}^k \in \mathcal{Y}}{\text{argmax}} \ \frac{\log p(\boldsymbol{y}^k \mid \boldsymbol{x})}{(|\boldsymbol{y}^k| - 1)^a} \tag{5}$$

where $\boldsymbol{y}^k = (y_0^k, \cdots, y_T^k)$. $|\boldsymbol{y}^k| - 1$ is used since the start token is not considered in calculating the length.

## 2.2 Attention Distribution

Attention distribution is a continuous vector whose element indicates the degree of attention paid to a source token. The element is formed by the accumulation of cross attention, i.e., $\sum_t \alpha_{t,i}$, where $\alpha_{t,i}$ refers to the cross attention that the $t_{th}$ token in the target sequence gives to the $i_{th}$ source token.[4] Specially, cross attention is a scaled dot-product [31] of hidden states of the source $\boldsymbol{x}$ and the target sequence $\boldsymbol{y}$. Notably, since Transformer-decoder is an auto-regressive model, the cross attention assigned by $t_{th}$ target token is actually calculated by $\boldsymbol{y}_{<t} = (y_0, \cdots, y_{t-1})$.

**Global Attention Distribution**. The global attention distribution $\boldsymbol{g} = [g_1, \cdots, g_i, \cdots, g_n] \in \mathbb{R}^n$ is the attention distribution given by the reference, where global attention $g_i$ refers to the total attention that the reference attends to the $i_{th}$ source token, and $n$ is the source length.

**Optimal Length**. The summation of $\boldsymbol{g}$, namely $\sum_{i=1}^n g_i$, is equal to $\sum_{i=1}^n \sum_{t=1}^T \alpha_{t,i} = T$ due to $\sum_{i=1}^n \alpha_{t,i} = 1$ in softmax operation, where $T$ is the reference length, or equivalently the optimal length $Z$.

**Local Attention Distribution**. The local attention distribution $\boldsymbol{l}_t^k = [l_{t,1}^k, \cdots, l_{t,i}^k, \cdots, l_{t,n}^k] \in \mathbb{R}^n$ is the attention distribution of the $k_{th}$ generated sequence and updated at each decoding step $t$. Thereinto, the local attention $l_{t,i}^k$ denotes the total attention paid to the $i_{th}$ source token by the $k_{th}$ beam sequence $(b_1^k, \cdots, b_t^k)$ and is dependent on the sequence generated before $t$, i.e., $\boldsymbol{b}_{t-1}^k = (b_0^k, \cdots, b_{t-1}^k)$.

# 3 Proposed Global-aware Inference

## 3.1 Global Scoring Mechanism

The global scoring mechanism consists of an attention scoring function and a length reward function. Given the global attention distribution $\boldsymbol{g}$, the attention scoring function $\mathcal{A}(\cdot)$ at the step $t$ depends on $\boldsymbol{b}_{t-1}^k$,

$$\mathcal{A}(\boldsymbol{b}_{t-1}^k) = \frac{\sum_{i=1}^n \min(l_{t,i}^k, g_i)}{\zeta_t^k}, \ \zeta_t^k = \sum_{i=1}^n l_{t,i}^k, \ t \geq 1 \tag{6}$$

where $\zeta_t^k$ indicates the total attention that the generated sequence $(b_1^k, \cdots, b_t^k)$ gives to the source, and $\zeta_t^k = |\boldsymbol{b}_{t-1}^k| = t$ because the assignable attention for each generated token is 1. Notably, Eq. 6 attains the maximum score provided that each $l_{t,i} \leq g_i$. As mentioned in Section § 1, the reason for

---

[4]Mean pooling is used for multi-layers and multi-heads

this design is that we desire the local attention $l_{T,i}$ at the termination is exactly $g_i$, since the final hypothesis is expected to attend to source tokens in the global-optimal manner. Meanwhile, we have $l_{T,i} > l_{t,i}$ for $t < T$ because $l_{t,i}$ monotonically increases on $t$ with $\alpha_{m,i}^l > 0$. Therefore, at any step, the local attention $l_{t,i}$ should not surpass $g_i$. Otherwise, the attention score will decline, and the penalty depends on the total amount by which these $l_{t,i}$ exceed $g_i$ (see Theorem 1). Further, the attention score could be considered as the proportion of correctly assigned attention to the total attention given by the generated sequence, where correct assignment indicates that all parts of $l_{t,i}$ do not exceed $g_i$. Also, it could be interpreted as the correct allocation probability of local attention against the global attention distribution (see Corollary 1.1). In this case, the total attention score can be expressed as the same multiplicative form as Eq. 1 to be elegantly integrated into beam search.

**Theorem 1.** *Let* $\mathcal{M} = \left\{ s : l_{t,s}^k > g_s \right\}$ *and* $\Delta_t^k = \left\{ \delta \mid \forall s \in \mathcal{M}, \ \delta = l_{t,s}^k - g_s \right\}$. *Given* $\Delta = \sum_{\delta \in \Delta_t^k} \delta$ *where* $\Delta \geq 0$, *then* $\mathcal{A}(\boldsymbol{b}_{t-1}^k)$ *decreases as* $\Delta$ *increases.*

**Corollary 1.1.** *The bound of* $\mathcal{A}(\boldsymbol{b}_{t-1}^k)$ *is between* $0$ *and* $1$.

*Proof.* See App. A. $\qquad\qquad\square$

In addition to the constraint for $l_{t,i}$, we still desire $l_{T,i} = g_i$ for each token. An ideal hypothesis should have two characteristics simultaneously. Namely, its attention score at the termination is the maximum 1, and its length equals the optimal length. Therefore, we introduce a length reward function to cooperate with the attention score to penalize the situation $l_{T,i} \neq g_i$, which will be discussed at the end of this subsection.

As mentioned before, the total attention score at the decoding step $t$ is defined as:

$$A(\boldsymbol{b}_{t-1}^k) = \prod_{m=1}^{t} \mathcal{A}(\boldsymbol{b}_{m-1}^k) \tag{7}$$

Thus, the joint scoring function $J(\cdot)$ is modified from Eq. 4:

$$
\begin{aligned}
J(\boldsymbol{b}_t, \boldsymbol{x}) &= \log p(\boldsymbol{b}_{t-1}^k | \boldsymbol{x}) + \beta \log A(\boldsymbol{b}_{t-1}^k) + \log p(v | \boldsymbol{x}, \boldsymbol{b}_{t-1}^k) \\
&= \sum_{m=1}^{t-1} \left( \log p(b_m | \boldsymbol{x}, (\boldsymbol{b}_{t-1}^k)_{<m}) + \beta \log \mathcal{A}(\boldsymbol{b}_{m-1}^k) \right) + \log p(v | \boldsymbol{x}, \boldsymbol{b}_{t-1}^k) + \beta \log \mathcal{A}(\boldsymbol{b}_{t-1}^k) \\
&= J(\boldsymbol{b}_{t-1}^k, \boldsymbol{x}) + \log p(v | \boldsymbol{x}, \boldsymbol{b}_{t-1}^k) + \beta \log \mathcal{A}(\boldsymbol{b}_{t-1}^k), \ t \geq 2
\end{aligned}
\tag{8}
$$

and $J(\boldsymbol{b}_1, \boldsymbol{x}) = \log p(v | \boldsymbol{x}, b_0^k) + \beta \log \mathcal{A}(b_0^k)$, where $\beta$ is a hyper-parameter to trade-off between the probability and attention score. Similar to $\log p(\boldsymbol{b}_t | \boldsymbol{x})$ in Eq. 4, $J(\boldsymbol{b}_t, \boldsymbol{x})$ is also an accumulative score. Consequently, at each step $t$, we only need compute $p(v | \boldsymbol{x}, \boldsymbol{b}_{t-1}^k)$ and $\mathcal{A}(\boldsymbol{b}_{t-1}^k)$. Compared with Eq. 4, the time complexity of each step is only increased by $\mathcal{O}(\mathcal{K})$ as there are $\mathcal{K}$ attention scores. Replacing $\log p(\boldsymbol{b}_t | \boldsymbol{x})$ in Eq. 3 by $J(\boldsymbol{b}_t, \boldsymbol{x})$, we can select the top $\mathcal{K}$ beams of each decoding step according to not only the probability distribution conditional on local information $\boldsymbol{b}_{t-1}^k$ but also the score conditional on global information $\boldsymbol{g}$.

Considering the length reward function, the final hypothesis is thus defined as:

$$\boldsymbol{y}^* = \operatorname*{argmax}_{\boldsymbol{y}^k \in \mathcal{Y}} \left( \frac{J(\boldsymbol{y}^k, \boldsymbol{x})}{|\boldsymbol{y}^k| - 1} + \beta \gamma R(\zeta_T^k, Z) \right) \tag{9}$$

where $R(\cdot)$ is the length reward function dependent on the optimal length $Z$ and the candidate hypothesis length $\zeta_T^k$. Exactly, $\zeta_T^k$ is the total attention that a candidate hypothesis $(y_1^k, \cdots, y_T^k)$ pays to the source and equals $|\boldsymbol{y}^k| - 1$. Besides, the attention score and length reward are weighted by a hyper-parameter $\gamma$, and the role of $\beta$ is to ensure that the two are at the same level relative to the probability. We remove $a$ in Eq. 5 as it only adjusts the length preference without really controlling the length.

The design of $R(\cdot)$ could be straightforward – one only need ensure that it increases as $\zeta_T^k$ approaches $Z$, and reaches the maximum only at $\zeta_T^k = Z$. In this paper, we design a step-wise length reward function $\mathcal{R}(\zeta_t^k, Z)$ to better balance the relationship between the attention score and the length reward and make the whole searching process as succinct as beam search. We put the design details of the step-wise length reward in App. B, and we regard Eq. 9 as the general scoring formulation of global-aware inference.

## 3.2 Predict the Global Attention Distribution

Since the reference is unknown practically, the global attention distribution could only be predicted from the source. We construct an attention-prediction model to learn the relationship between the source tokens and the global attention distribution.

The input of the attention-prediction model is the fixed encoder output $\boldsymbol{E} \in \mathbb{R}^{n \times d}$ of BART or PEGASUS plus learnable positional encodings $\boldsymbol{P} \in \mathbb{R}^{n \times d}$, where $d$ is the dimension of hidden states. The input is fed to a learnable Transformer-encoder to obtain $\widetilde{\boldsymbol{E}} \in \mathbb{R}^{n \times d}$ that is encoded with additional context information, followed by a linear transformation with an exponential function:

$$\widehat{\boldsymbol{g}} = \exp\left(\widetilde{\boldsymbol{E}}\boldsymbol{W}_g + \boldsymbol{b}_g\right) \tag{10}$$

where $\widehat{\boldsymbol{g}} \in \mathbb{R}^n$ refers to the prediction of $\boldsymbol{g}$, $\boldsymbol{W}_g \in \mathbb{R}^{d \times 1}$ and $\boldsymbol{b}_g \in \mathbb{R}^n$ are the learnable weights and biases. The exponential function is imposed to ensure $\widehat{\boldsymbol{g}} > 0$. We choose the exponential function for this operation because it is shown stable in the training and testing stage. Given the objective of minimizing the distance between $\widehat{\boldsymbol{g}}$ and $\boldsymbol{g}$, the loss is defined as their Euclidean distance:

$$\mathcal{L} = \|\widehat{\boldsymbol{g}} - \boldsymbol{g}\|_2 \tag{11}$$

The predicted optimal length $\widehat{Z}$ is the sum of elements in $\widehat{\boldsymbol{g}}$. Note that the length reward function is not affected no matter whether $\widehat{Z}$ is an integer or not.

# 4 Experiment

## 4.1 Setup

**Datasets**. We evaluate the performance on totally 9 summarization datasets, where 2 datasets (CNN/DM [10], XSUM [5]) with BART [17] and 8 datasets (XSUM [5], BillSum [15], Multi-News [6], NewsRoom [9], WikiHow [16], Reddit TIFU [34], arXiv and PubMed [2]) with PEGASUS [39]. Thereinto, XSUM [5] is a highly abstractive dataset whose summaries are all expressed in a short sentence.

**Implementation Details**. We adopt a randomly initialized 2-layer transformer-encoder in the attention-prediction model wherethe structure of each layer is the same as the BART-encoder layer. The optimizer is the Adabelief-optimizer [41] with eps $1e - 16$, betas $(0.9, 0.999)$, weight decay $1e - 4$ and learning rate $2e - 5$. The attention-prediction model is trained on the training set for about $50,000$ steps, and checkpoints are saved per $10,000$ steps to select the best checkpoints on the development set. Since the attention prediction is slightly different from common sequence tagging tasks, we have summarized two notable points after several attempts – the dropout rate should be $0$, and a small learning rate is preferred. All experiments are conducted on 3 *RTX 6000*. We include the global-aware inference in the generation code of *HuggingFace transformers* [36]. At the time of evaluation, ROUGE-1, ROUGE-2 & ROUGE-L (R-1, R-2 & R-L) scores [22] are computed from the ROUGE code[5] used by BART [17].

**Hyper-parameter Selection**. Although the global-aware inference requires two new hyper-parameters $\gamma$ and $\beta$, some original hyper-parameters of beam search, namely length penalty, minimum and maximum length, are omitted. The searching scopes of $\beta$ and $\gamma$ are in $\{2, 4, 6, 10, 12, 15, 18, 20\}$ and $\{0, 0.5, 1, 1.5, 2\}$, respectively. According to the numerical tests on the development set, we finally choose $\beta = 12$, $\gamma = 1.5$ for CNN/DM and $\beta = 4$, $\gamma = 0$ for XSUM. As limited improvement could be observed from larger $\gamma$'s, we recommend $\gamma = 1$ for normal or longer targets. When testing the global-aware inference with PEGASUS [39], we directly use **empirical hyper-parameters** for each dataset, namely $\beta = 4$, $\gamma = 0$ for XSUM and $\beta = 12$, $\gamma = 1$ for other 7 datasets. The beam size $\mathcal{K}$ follows the setups in BART [17] and PEGASUS [39].

## 4.2 Results

**Comparison with Beam Search**. Beam search is a hard-to-beat baseline which has stood the test of time and proven its superiority in practice for long [24]. In Table 2, we compare our global-aware

---

[5]https://github.com/pltrdy/files2rouge

Table 2: ROUGE $F_1$ scores of summaries generated by global-aware, in comparison to beam search with length regularizations. Notably, global-aware uses empirical hyper-parameters.

| $\mathcal{K} = 8$ | XSUM | | | BillSum | | | Multi-News | | | WikiHow | | |
|---|---|---|---|---|---|---|---|---|---|---|---|---|
| | R-1 | R-2 | R-L | R-1 | R-2 | R-L | R-1 | R-2 | R-L | R-1 | R-2 | R-L |
| **Beam search** [39] | 47.05 | 24.53 | 39.33 | 57.00 | 39.65 | 52.70 | 47.29 | 18.91 | 43.31 | 41.86 | 19.04 | 40.40 |
| **Global-aware** | **47.33** | **24.66** | **39.50** | **58.66** | **40.12** | **53.96** | **47.95** | **19.08** | **43.93** | **42.82** | **19.68** | **41.43** |

| | Reddit TIFU | | | NewsRoom | | | PubMed | | | arXiv | | |
|---|---|---|---|---|---|---|---|---|---|---|---|---|
| | R-1 | R-2 | R-L | R-1 | R-2 | R-L | R-1 | R-2 | R-L | R-1 | R-2 | R-L |
| **Beam search** [39] | 27.55 | 8.67 | 22.12 | 42.05 | 29.88 | 38.70 | 44.25 | 19.19 | 41.11 | 43.82 | 16.75 | 39.28 |
| **Global-aware** | **28.31** | **9.13** | **23.30** | **44.68** | **31.71** | **41.28** | **45.78** | **20.16** | **42.62** | **44.92** | **17.41** | **40.31** |

inference to beam search with length regularizations (i.e., $\alpha$ in Eq. 5, accompanied with two hard constraints, namely minimum length and maximum length). We strictly follow the hyper-parameter setups of PEGASUS [39] in terms of beam search, while we only adopt empirical hyper-parameters for our method. Even so, significant improvements can be observed on all the data sets, especially when the summary is of normal or longer length.

Table 3: Comparison with other methods.

| $\mathcal{K} = 4$ | CNN/DM | | |
|---|---|---|---|
| | R-1 | R-2 | R-L |
| **Beam search** [17] | 44.12 | 21.21 | 40.89 |
| **+ Our coverage** | 44.74 | 21.69 | 41.48 |
| **+ Repetition penalty** [13] | 44.11 | 21.14 | 40.87 |
| **+ Attention masking** [1] | **45.54** | **22.24** | **42.44** |
| **Global-aware** | 45.13 | 21.77 | 42.04 |

| $\mathcal{K} = 6$ | XSUM | | |
|---|---|---|---|
| | R-1 | R-2 | R-L |
| **Beam search** [17] | 45.38 | 22.32 | 37.15 |
| **+ Our coverage** | 44.54 | 21.82 | 36.97 |
| **+ Repetition penalty** [13] | 45.40 | 22.31 | 37.13 |
| **+ Attention masking** [1] | 45.35 | 22.31 | 37.15 |
| **Global-aware** | **45.57** | **22.60** | **37.61** |

Table 4: ORACLE and ablation results.

| | CNN/DM | | |
|---|---|---|---|
| | R-1 | R-2 | R-L |
| **ORACLE global-aware** | **51.85** | **28.13** | **48.68** |
| *-w/o length reward* | 50.46 | 27.53 | 47.43 |
| **Global-aware** | **45.13** | **21.77** | **42.04** |
| *-w/o length reward* | 44.39 | 21.58 | 41.41 |
| *-w/o attention score* | 44.12 | 21.29 | 40.91 |

| | XSUM | | |
|---|---|---|---|
| | R-1 | R-2 | R-L |
| **ORACLE global-aware** | **49.50** | 26.24 | 41.13 |
| *-w/o length reward* | 48.92 | **26.48** | **41.45** |
| **Global-aware** ($\gamma = 1$) | 45.44 | 22.15 | 37.11 |
| *-w/o length reward* | **45.57** | **22.60** | **37.61** |
| *-w/o attention score* | 45.23 | 21.88 | 36.73 |

Table 5: Improvements of attention head masking and global-aware on beam search [39] in terms of ROUGE-L $F_1$ score. Both use empirical setups.

| | XSUM | BillSum | Multi-News | WikiHow | Reddit | NewsRoom | PubMed | arXiv |
|---|---|---|---|---|---|---|---|---|
| **Attention head masking** [1] | -0.31 | 0.23 | 0.34 | 0.10 | -0.16 | 1.24 | 0.35 | 0.21 |
| **Global-aware** | **0.17** | **1.26** | **0.62** | **1.03** | **1.18** | **2.58** | **1.51** | **1.03** |

**Comparison with Other Attention Approaches**. In this part, we focus on comparing other approaches which also exploit certain attention distributions to improve beam search. The first is the coverage penalty [37, 21]. To enhance its performance in summarization, we replace its preset attention distribution with our predicted global attention distribution. Note that the coverage function can only evaluate the generated sentences at the terminal step. Instead of comparing the global-aware inference to the methods [29, 20, 7] that aim to reduce repetition using the dynamic attention distribution, we compare our algorithm with the CTRL repetition penalty [13] which has similar motivation but is more systematic and independent of training. Table 3 lists the comparison results against different algorithms. It can be observed that our global-aware approach can improve the performance of beam search stably and fundamentally. We also observe that the attention head masking [1] appears to outperform the global-aware approach on CNN/DM, but it fails to gain any improvement on XSUM. To further show the advantage of the proposed approach, we will take a closer examination on the attention head masking [1] and our proposed approach in the next part.

**Further Comparison with Attention Head Masking**. First, one should bear in mind that the attention head masking [1] acts on the model instead of beam search, which is opposite to us. Specifically, it selects contents during decoding by disabling partial attention heads for unimportant

Table 6: Generate CNN/DM summaries with XSUM's style.

| R1/R2/RL | Shorter beam search | Global-aware† |
|---|---|---|
| $F_1$ score | **43.6/20.9**/40.4 | **43.6**/20.4/**40.6** |
| Recall | **48.9/23.4/45.2** | 40.4/18.8/37.6 |
| Precision | 41.4/19.9/38.3 | **50.5/23.8/47.0** |

Table 7: Generate summaries with corrupted attention distributions.

| | R-1 | R-2 | R-L |
|---|---|---|---|
| Beam search | 27.00 | 12.21 | 23.88 |
| Global-aware (10K) | 28.78 | 12.82 | 25.51 |
| Global-aware (100K) | **29.59** | **13.72** | **26.30** |

Figure 2: Sensitive analysis in the test set.

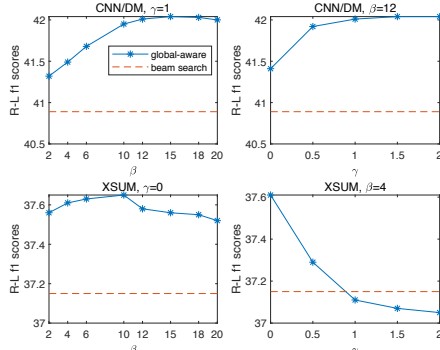

tokens to decrease the attention to these tokens. According to the reported results presented in Table 3, we can see that although attention masking achieves amazing results on CNN/DM, it does fail completely on XSUM. Since hiding unimportant tokens from some heads results in the loss of context information of salient tokens, this would lead to its instability. Thus, it could be ineffective for tasks that require contextual information of the whole source such as XSUM. Taking a further comparison, we deploy the attention head combinations selected for CNN/DM and XSUM to examine its effect on PEGASUS [39]. These comparison results are shown in Table 5. Evidently, our method enjoys a robust feature that is able to boost summary inference on various datasets and models even with the same set of hyper-parameters. In contrast, attention masking [1] behaves much sensitive to the changes of models and datasets. Besides, attention masking has to construct its training saliency labels based on the longest common subsequences between a reference summary and the source. This may be hardly achieved in some text generation tasks (e.g. translation) where no common subsequence exists at all. Such drawback presents one main limitation for attention masking.

**ORACLE Results and Ablation Study**. ORACLE refers to the global-aware inference combined with (true) global attention distribution instead of predicted global attention distribution. The related results have been presented in Table 4, and the significant boosting certifies that the proposed method could improve beam search with the global attention distribution. On the other hand, we conduct ablation experiments on ORACLE and global-aware. Both results indicate that length reward plays an important role in generating normal-length text but causes adverse effect on generating text of very short length. Besides, the performance declines significantly when only length reward is applied, due to the fact that sole length reward cannot calibrate beam search step-wise.

**Robustness**. We intend to examine the robustness of our proposed global-aware algorithm in this and next part. To do so, we substitute the parameters in CNN/DM's attention prediction model to XSUM's to create more abstractive summaries for CNN/DM. For comparison, we set the minimum length of beam search as 0 to allow it to generate shorter summaries. Table 6 shows the $F_1$ score, recall and precision of the shorter beam search and global-aware†. It is surprising that even by using the attention distribution from a different dataset with distinct properties, the proposed global-aware mechanism still manages to generate meaningful summaries with competitive $F_1$ scores, proving the robustness of this algorithm. Moreover, the higher Precision and lower Recall of the global-aware suggest that although information is partially lost, the algorithm still summarizes core information in a concise format, compared to the standard beam search. On the other hand, we exploit a BART model fine-tuned on CNN/DM to generate summaries of NewsRoom directly, and the ROUGE scores of beam search are shown in Table 7. Next, we randomly select 10K and 100K samples from the training set and use them to fine-tune the attention-prediction model, where the global-aware improves beam search substantially. The experiment once again validates the robustness of the proposed inference algorithm, as it maintains a reasonably good performance even from learning a corrupted attention distribution from a BART without fine-tuning.

**Sensitive Analysis**. We further examine the performance robustness with respect to the hyper-parameters. From Figure 2, we could see the global-aware inference is always better than beam search in CNN/DM, no matter how its hyper-parameters change. Besides, the performance is less sensitive to the hyper-parameters when $\beta \geq 10$ or $\gamma \geq 1$. While in XSUM, the global-aware could improve beam search stably with $\gamma = 0$, but there is a significant decline when applying length

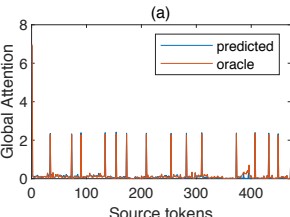 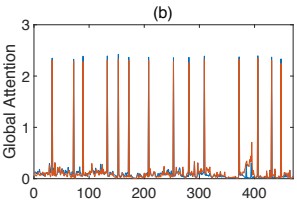 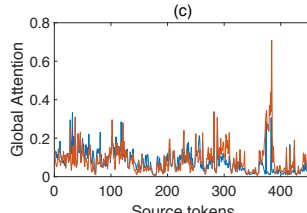

Figure 3: Predicted and ORACLE global attention in BART. There are attention distributions of (a) the whole source, (b) the source without the start & end tokens, (c) the source without the start & end tokens and full stops.

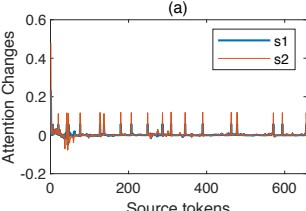 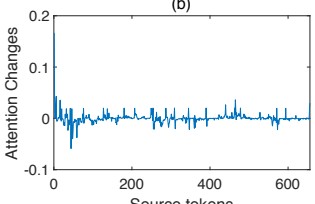 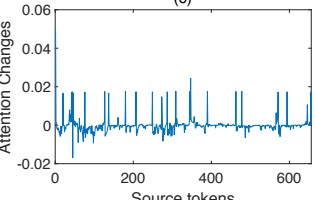

Figure 4: Changes of the attention distribution when (a) one word in the reference is replaced by a similar word (s1) and a random word (s2), (b) the sentence order of the reference is shuffled, (c) a factual knowledge in the reference is distorted.

reward. In fact, the attention score favors shorter hypotheses, and the length reward could alleviate the bias. However, if the references of a dataset are already very short such as XSUM, the length reward may lead to a counterproductive effect. Since the setup of CNN/DM is applicable to most datasets, we argue that the global-aware inference is robust to both hyper-parameters in most cases.

## 5 Analysis on Global Attention Distribution

### 5.1 Distribution Law

The distribution law of global attention in BART [17] is shown in Figure 3. It is observed that most attention is assigned to the start token and full stops which are not semantically salient, and most of the remaining attention is unevenly allocated to (semantically) overlapped tokens between the source and the reference (i.e., important words). It is worth mentioning that the importance here is no longer a simple binary classification like in [7, 1], but a continuous numerical value decided by the model knowledge learned from data. In general, one should not simply equate the global attention with the word importance, but should be clear that it essentially reflects knowledge learned by the attention model such as the syntactic and semantic structure of sentences. Meanwhile, the distribution law indicates that attention distributions in pre-trained models may not be relevant with the uniform distribution at all. That is to say, it is not reasonable to still use an uniform attention threshold (like the threshold 1 preset in [37, 7]) to regulate the decoding of pre-trained models. Last but not least, our general motivation is to alleviate locality bias by aligning the attention distribution of reference and hypothesis, which does not really care how the global attention is distributed only if it is predictable. However, the proposed penalty mechanism is indeed insensitive to some distributions, and we will provide more thinking about this in App. G by applying the global-aware inference to translation.

### 5.2 Why It can be Predicted from Source?

Since the ORACLE experiments indicated that the global attention is helpful for decoding, the concern remains if it is predictable using only source. In App. E.1 we will show the predictability empirically, while here we just provide an interesting explanation. In our opinion, the global attention distribution is an interpretable representation of reference which not only has the characteristics of hidden representation but also can be explained by the source tokens. First of all, given the source, the global attention is calculated by the input reference and trained neural network parameters;

this is similar to achieving hidden representation. Moreover, like hidden representation, the global attention distribution could also capture the semantic similarity, e.g., replacing a reference word with a semantically similar word leads to slighter attention changes than that with a random word (see Figure 4.a). Besides, it is observed from Figure 4.b and Figure 4.c that global attention is able to represent the sentence order and factual knowledge. On the other hand, the global attention distribution enjoys a distinct characteristic that each of its features can be explained as the degree of attention paid to a source token, which means changes of such representation are regular to some extent. For example, in Figure 4.c, we distort a numerical number in the reference to violate the fact stated in the source, and then find that the attention assigned to the actual number originally is most transferred to the special tokens and punctuation. Overall, similar sources should have similar global attention distributions, since similar sources often have similar references and global attention distribution is a representation of reference. Moreover, the global attention and source tokens are in an one-to-one correspondence. Thereby, we argue that it is convenient to use the source to predict the global attention distribution.

## 6 Related Work

In the field of text generation, efforts have been made to boost beam search with regards to the attention distribution. For instance, some studies engage the attention distribution to penalize the situation where sources fail to be fully covered in translation tasks [37, 21], while others [7, 29, 20] incorporate dynamic attention distributions to evade tokens that have been highly regarded to reduce repetition. However, none of the aforementioned studies attempts to apply the global attention distribution to acquire the knowledge that the level of attention should be allocated to each token from the reference. Further, the existing score functions used by those studies are rather different from the proposed global scoring mechanism. More details can be seen in App. D.

In addition to the attention distribution, other techniques are developed to improve beam search in terms of length bias [38], diversity-less [32], vapidity [11] and degradation [25, 3]. These methods are not included for comparison because they are not suitable for summarization [38, 11], or do not aim to enhance beam search as the main purpose [32, 25, 3]. Besides, we argue that these patches to beam search are supposed to be hard for improving the performance stably and fundamentally (as shown in Table 3 given by our global-aware method) because they fail to specify what a final hypothesis should look like and are easy to trap into local optimums.

## 7 Conclusion

Beam search tends to fall into local optimums due to the lack of global information during inference. To calibrate beam search on the premise of global awareness, this study proposes a predictable protocol to stipulate how a global optimal hypothesis should attend to source tokens. By training a simple prediction model of the global attention distribution, a novel global scoring mechanism is then developed to regulate the inference on a step-wise basis. Our experiment shows that the proposed global-aware beam search generates higher-quality summaries, relative to beam search and other counterparts of the similar nature. Besides, the proposed algorithm is proven robust in generating meaningful texts even with corrupted attention distributions, implying its potential to cooperate with user-defined global attention distributions. We plan to focus our future study on generalizing the global-aware inference to a broader range of text generation tasks, including not only text2text but also image caption [33], multimodal summarization [19], graph2text [14] and data2text [26].

### Acknowledgments and Disclosure of Funding

Funding in direct support of this work: XJTLU Key Programme Special Fund KSF-A-14 and KSF-P-02, National Natural Science Foundation of China under No. 61876155 and Jiangsu Science and Technology Programme under No. BE2020006-4.

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
