&= \sum_{i=1}^{n} \min\left( \frac{l_{t,i}^k}{\zeta_t^k}, \frac{g_i}{\zeta_t^k} \right) \\
&= \sum_{i=1}^{n} \min\left( 0, \frac{g_i - l_{t,i}^k}{\zeta_t^k} \right) + \sum_{i=1}^{n} \frac{l_{t,i}^k}{\zeta_t^k}
\end{aligned}
\tag{12}
$$

Since $\zeta_t^k = \sum_{i=1}^{n} l_{t,i}^k$, we have $\sum_{i=1}^{n} \frac{l_{t,i}^k}{\zeta_t^k} = 1$. Besides, with $\zeta_t^k > 0$,

$$
\forall s \in \mathcal{M}, \, \min\left( 0, \frac{g_s - l_{t,s}^k}{\zeta_t^k} \right) = \frac{g_s - l_{t,s}^k}{\zeta_t^k} = \frac{-\delta}{\zeta_t^k}
\tag{13}
$$

Thus Eq. 12 is equal to:

$$
\begin{aligned}
\mathcal{A}(\boldsymbol{b}_{t-1}^k) &= 0 + \sum_{\delta \in \Delta_t^k} \frac{-\delta}{\zeta_t^k} + 1 \\
&= 1 - \frac{\Delta}{\zeta_t^k}
\end{aligned}
\tag{14}
$$

Then it is easy to prove that $\mathcal{A}(\boldsymbol{b}_{t-1}^k)$ goes down as $\Delta$ goes up. $\qquad \square$

**Corollary 1.1.** *The bound of $\mathcal{A}(\boldsymbol{b}_{t-1}^k)$ is between $0$ and $1$.*

*Proof.* Since we have achieved $\mathcal{A}(\boldsymbol{b}_{t-1}^k) = 1 - \frac{\Delta}{\zeta_t^k}$ where $\mathcal{A}(\boldsymbol{b}_{t-1}^k)$ decreases monotonically on $\Delta$ and $\Delta \geq 0$, $\mathcal{A}(\boldsymbol{b}_{t-1}^k)$ reaches the maximum $1$ when $\Delta = 0$. Next, we assume a beam can be generated indefinitely. In this case, there must be a situation that each $l_{t,i}^k > g_i$. Therefore we have

$$
\begin{aligned}
\mathcal{A}(\boldsymbol{b}_{t-1}^k) &= 1 - \frac{\sum_{i=1}^{n}(l_{t,i}^k - g_i)}{\zeta_t^k} \\
&= 1 - \frac{\sum_{i=1}^{n} l_{t,i}^k - \sum_{i=1}^{n} g_i}{\zeta_t^k} \\
&= 1 - \frac{\zeta_t^k - \sum_{i=1}^{n} g_i}{\zeta_t^k} \\
&= \frac{\sum_{i=1}^{n} g_i}{\zeta_t^k}
\end{aligned}
\tag{15}
$$

where $\lim_{\zeta_t^k \to \infty} \mathcal{A}(\boldsymbol{b}_{t-1}^k) = 0$. Therefore, the bound of $\mathcal{A}(\boldsymbol{b}_{t-1}^k)$ is proven to be between $0$ and $1$. $\quad \square$

# B  Length Reward

In the first subsection, we will introduce our step-wise length reward function and how it works. In the second part, we will discuss in details how it is designed.

## B.1  Our Step-wise Length Reward

Our length reward function $\mathcal{R}(\zeta_t^k, Z)$ is to give a length score to a beam at each decoding step, so that its cumulative score $R(\zeta_T^k, Z)$ for a terminated beam could approach the maximum as its length $\zeta_T^k$ gets closer to the optimal length $Z$. Our designed length reward function is as follows:

$$\mathcal{R}(\zeta_t^k, Z) = -\frac{\left|\zeta_t^k - \frac{Z}{\sqrt{2}} - 0.5\right|}{Z} \tag{16}$$

Eq. 8 is thus expanded to:

$$J(\boldsymbol{b}_t, \boldsymbol{x}) = \log p(\boldsymbol{b}_{t-1}^k|\boldsymbol{x}) + \beta\left(\log A(\boldsymbol{b}_{t-1}^k) + \gamma \sum_{m=1}^{t} \mathcal{R}(\zeta_m^k, Z)\right) + \log p(v|\boldsymbol{x}, \boldsymbol{b}_{t-1}^k) \tag{17}$$

$$= J(\boldsymbol{b}_{t-1}^k, \boldsymbol{x}) + \log p(v|\boldsymbol{x}, \boldsymbol{b}_{t-1}^k) + \beta\left(\log \mathcal{A}(\boldsymbol{b}_{t-1}^k) + \gamma\mathcal{R}(\zeta_t^k, Z)\right), \ t \geq 2$$

In this case, the final score of $\boldsymbol{y}^k$ is re-defined as

$$\boldsymbol{y}^* = \underset{\boldsymbol{y}^k \in \mathcal{Y}}{\operatorname{argmax}} \frac{J(\boldsymbol{y}^k, \boldsymbol{x})}{|\boldsymbol{y}^k| - 1} \tag{18}$$

The original $R(\zeta_T^k, Z)$ in Eq. 9 has been integrated to $J(\boldsymbol{y}^k, \boldsymbol{x})$, and exactly equals $\frac{\sum_{t=1}^{T} \mathcal{R}(\zeta_t^k, Z)}{|\boldsymbol{y}^k| - 1} = \frac{\sum_{t=1}^{\zeta_T^k} \mathcal{R}(t, Z)}{\zeta_T^k}$ which gets the maximum at $\zeta_T^k = Z$. Related theorem and proof are presented in the next subsection.

## B.2  How It Is Designed

Intuitively, if we want to give a score to the beam length at each step, the score should reach the maximum when the beam length gets the optimality. Accordingly it can be defined as:

$$\mathcal{R}(\zeta_t^k, Z) = -\frac{\left|\zeta_t^k - Z\right|}{Z}, \ Z = \sum_{i=1}^{n} g_i \tag{19}$$

Since $\zeta_t^k = t$, it is represented as:

$$\mathcal{R}_t = -\frac{|t - Z|}{Z} \tag{20}$$

Then, we test whether its cumulative value at the terminated step $R(\zeta_T^k, Z)$ reaches the maximum when $\zeta_T^k = Z$. The cumulative length reward is defined as

$$R(\zeta_T^k, Z) = \frac{\sum_{t=1}^{\zeta_T^k} \mathcal{R}_t}{\zeta_T^k} \tag{21}$$

If we replace $\zeta_T^k$ by $j$, it can be represented as:

$$R_j = \frac{\sum_{t=1}^{j} \mathcal{R}_t}{j} \tag{22}$$

Unfortunately, when we plot the curve of $R_j$ as the orange dummy line in Figure 5, we find that it reaches the maximum at a length longer than the optimal length. This prompts us to translate $\mathcal{R}_t$ from the green dummy line to the green full line, and the adjusted length reward is designed as:

$$\mathcal{R}_t = -\frac{\left|t - \frac{Z}{\sqrt{2}} - 0.5\right|}{Z} \tag{23}$$

which is the same as Eq. 16. After the adjustment, the normalized cumulative length reward $R_j$ (the orange solid line in Figure 5) peaks right at the optimal length.

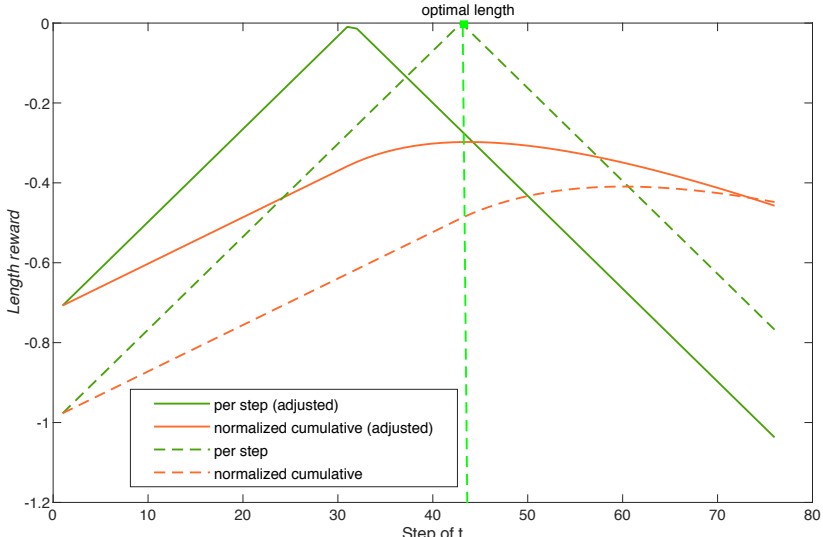

Figure 5: Difference between length reward and adjusted length reward.

**Theorem 2.** *Given $\mathcal{R}_t = -\frac{|t+D-Z|}{Z}$ ($t \geq 1$) and $R_j = \frac{\sum_{t=1}^{j} \mathcal{R}_t}{j}$ reaches the maximum when $j = Z$, then $D$ is approximate to $-\frac{Z}{\sqrt{2}} - 0.5 + Z$.*

*Proof.* Since $t \geq 1$ and $t \in \mathbb{Z}^+$, $\mathcal{R}_t$ can be regarded as two arithmetic sequences with difference of $\frac{1}{Z}$ and $-\frac{1}{Z}$, so that it is easy to calculate $\sum_{t=1}^{j} \mathcal{R}_t$. Then $R_j$ could be computed by:

$$R_j = -\frac{1}{j} \left[ \frac{(Z-D-1)(Z-D)}{2Z} + \frac{(j+D-Z)(j+D-Z+1)}{2Z} \right] \tag{24}$$

Hence the derivative of $R_j^a$ is:

$$\frac{\partial R_j}{\partial j} = \frac{\frac{2(Z-D)^2 - 2(Z-D)}{j^2} - 1}{2Z} \tag{25}$$

In addition, when $j = Z$, letting $\frac{\partial R_j}{\partial j} = 0$, we have:

$$2D^2 - (4Z-2)D + Z(Z-2) = 0 \tag{26}$$

If we solve this equation, we can get:

$$D = \frac{4Z - 2 - \sqrt{8Z^2 + 4}}{4} \tag{27}$$

Omitting the constant item in the radical due to $8Z^2 \gg 4$ under normal conditions, we can get:

$$D = Z - 0.5 - \frac{Z}{\sqrt{2}} \tag{28}$$

If we further take $D$ into $\mathcal{R}_t$, we can get Eq. 23. $\qquad \square$

## C   Flow Chart: Global-aware vs. Beam Search

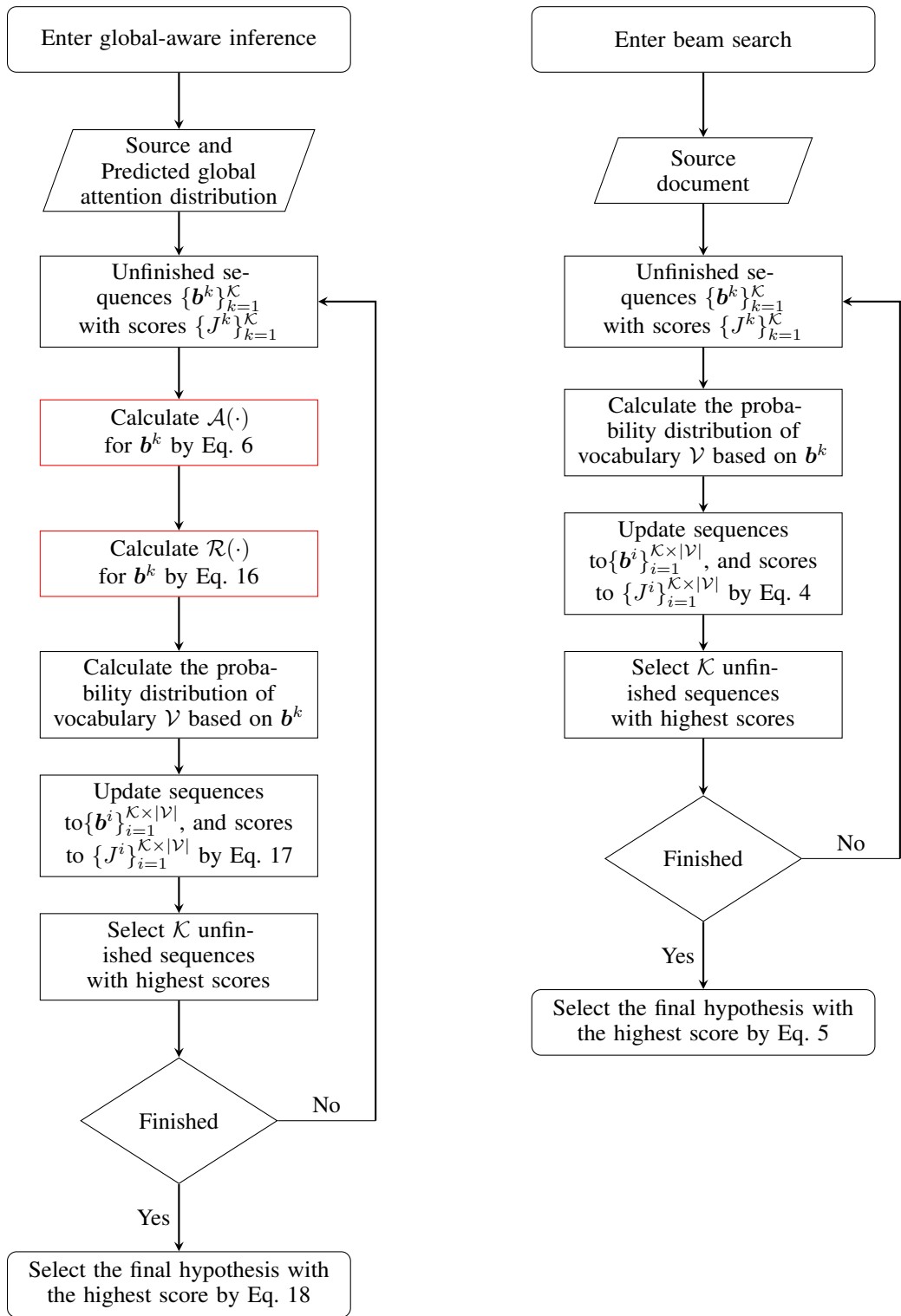

# D  Scoring Mechanism Comparison

Table 8: Scoring Mechanism Comparison. Training: whether the score function is a part of loss function of the seq-to-seq model. Inference: whether the score function is a part of the hypothesis score. Desired Situation: the score function reaches the optimum in this situation. †: the function described in the paper [7] is only activated at the termination, but their code added a step-wise modification on it.

| Article | Score Function | Training | Inference | Step-wise | Desired Situation |
|---------|----------------|----------|-----------|-----------|-------------------|
| [37] | $\sum_{i=1}^{n} \log \min \left( \sum_{t=1}^{T} \alpha_{t,i}, 1 \right)$ | ✗ | ✓ | ✗ | $\sum_{t=1}^{T} \alpha_{t,i} \geq 1$ |
| [21] | $\sum_{i=1}^{n} \log \min \left( \sum_{t=1}^{T} \alpha_{t,i}, \beta \right)$ | ✗ | ✓ | ✗ | $\sum_{t=1}^{T} \alpha_{t,i} \geq \beta$ |
| [29] | $-\sum_{i=1}^{n} \min \left( \alpha_{t,i}, \sum_{m=1}^{t-1} \alpha_{m,i} \right)$ | ✓ | ✗ | ✓ | $\sum_{m=1}^{t-1} \alpha_{m,i} < \alpha_{t,i}$ |
| [20] | $1 - \sum_{i=1}^{n} \min \left( \alpha_{t,i}, \sum_{m=1}^{t-1} \alpha_{m,i} \right)$ | ✓ | ✓ | ✓ | $\sum_{m=1}^{t-1} \alpha_{m,i} < \alpha_{t,i}$ |
| [7] | $n - \sum_{i=1}^{n} \max \left( \sum_{m=1}^{t} \alpha_{t,i}, 1 \right)$ | ✗ | ✓ | ✓† | $\sum_{m=1}^{t} \alpha_{m,i} \leq 1$ |
| Ours | $\frac{\sum_{i=1}^{n} \min\left(\sum_{m=1}^{t} \alpha_{m,i}, g_i\right)}{\sum_{i=1}^{n} \sum_{m=1}^{t} \alpha_{m,i}} + \gamma \mathcal{R}(\zeta_t^k, Z)$ | ✗ | ✓ | ✓ | $\sum_{m=1}^{t} \alpha_{m,i} < g_i, t < T$ **and** $\sum_{m=1}^{T} \alpha_{m,i} = g_i$ |

We summarize the characteristics of each score function in Table 8, where $\alpha_{t,i}$ refers to the cross attention that the $t_{th}$ generated token pays to the $i_{th}$ source token. Previous works [37, 21, 29, 20, 7] design the score function based on some assumptions which may be invalid. Specifically, [37], [21] and [7] use the same constant (1 or $\beta$) to constrain all source tokens, based on the assumption that the minimum/maximum attention allocated to each token is the same. [29] and [20] assume the attention assigned before $t$ should be lower than the attention of $t$. Obviously, both assumptions may often not hold in practice. By contrast, our global scoring mechanism does not depend on any assumption but follows the nature of generation, i.e., the growing local attention should not surpass the global attention during inference and exactly reach it at the termination.

# E   Discussions

## E.1   Predictability of Global Attention

We plot the decline trend of training loss and validation loss on CNN/DM and XSUM to Figure 6. It is observed that training loss has been falling significantly, while validation loss has an upward trend at later epochs.

On the other hand, we randomly select $1,000$ test samples from each dataset and plot their average $R^2$ scores of the predicted optimal attention distributions in Figure 7. For each sample, its $R^2$ score is equal to $1 - \frac{\sum_{i=1}^{n}(o_i - \hat{o}_i)^2}{\sum_{i=1}^{n}(o_i - \bar{o})^2}$, where $R^2 \in [-\infty, 1]$. A higher $R^2$ signifies better model fitting, while $R^2 < 0$ means the model is entirely invalid.

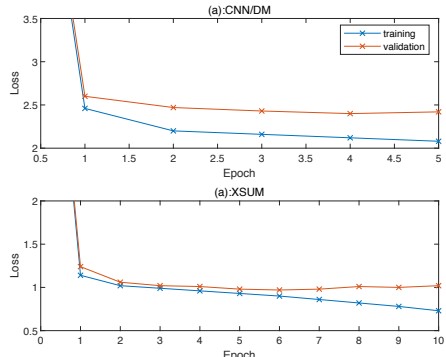

Figure 6: Loss trend.

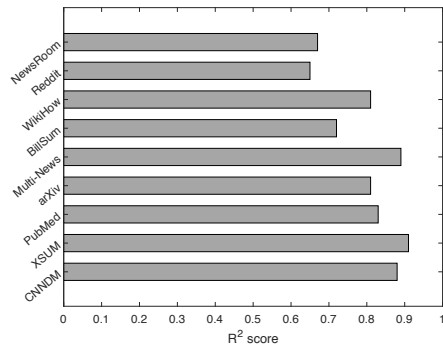

Figure 7: Fitting of att-prediction model.

## E.2   Degradation of Beam Search

It is widely known that text quality generally degrades with increased beam size [3, 25, 24]. The degradation originates from the gap between training and inference, where teacher forcing training only need ensure the local optimality, but beam search pursues the global optimality [35, 28, 40]. The proposed algorithm reduces the gap by informing beam search how the global optimal hypothesis attends to the source so that beam search can be calibrated to pursue a more reasonable global optimum instead of that obtained by teacher forcing training. We enlarge the beam size and find that the global attention distribution can guide beam search to overcome degradation, though the practical improvement might not be significant due to the prediction bias. According to Figure 8, the average ROUGE $F_1$ score of beam search summaries declines sharply as beam size increases. Even if there is a prediction bias, the global-aware inference manages to resist degradation effectively with increased beam sizes. Moreover, a significant boosting in ROUGE scores is observed as beam size increases in the ORACLE global-aware inference, where the true potential of the algorithm is reflected.

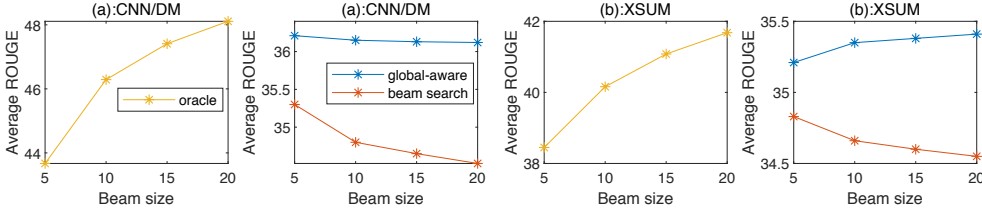

Figure 8: Degradation of beam search.

## E.3   Length Comparison

As shown in Table 9, even with empirical hyper-parameters, our method often leads to more concise summary with its length closer to the reference length on many datasets. By contrast, beam search,

Table 9: The average summary length. †: consisting of shorter beam search and global-aware†. *: without length reward. In addition to the two datasets, the numbers closer to the reference length are bolded while shorter lengths are underlined. All reference summaries are truncated to 256 tokens.

| Token Length | CNNDM | Bill | Multi | Wiki | Reddit | NRoom | Pub | arXiv | CNNDM† | XSUM* |
|---|---|---|---|---|---|---|---|---|---|---|
| *reference* | 67.5 | 176.5 | 232.9 | 70.5 | 26.2 | 34.4 | 214.7 | 191.3 | 67.5 | 26.1 |
| *beam search* | 82.3 | 162.5 | 219.4 | **49.8** | 36.5 | 46.7 | **205.8** | **176.1** | 78.2 | 21.8 |
| *global-aware* | **64.9** | **165.6** | **223.7** | 46.2 | **20.8** | **31.2** | 200.4 | 166.4 | 53.2 | 22.2 |

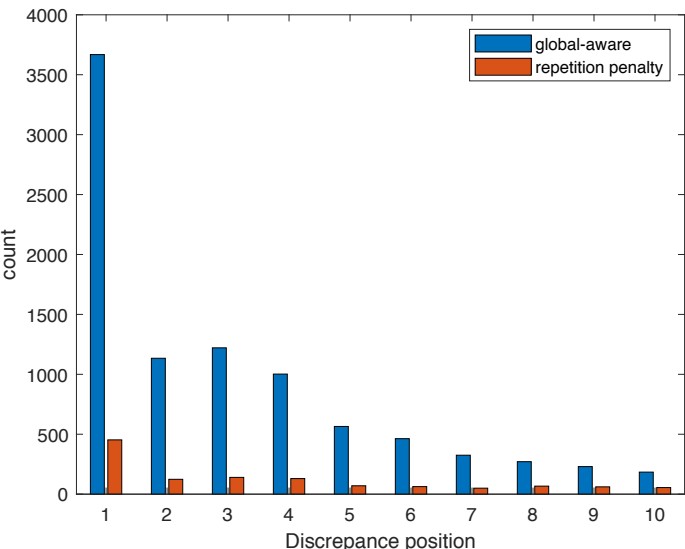

Figure 9: The position that the summary starts to differ from the beam search summary.

which conducts searching for length penalty and length constraints, generates many lengthy summaries on some datasets. For example, In CNN/DM, Reddit, and NewsRoom datasets, beam search must extend summaries to obtain higher ROUGE scores, making the generated text more bloated. Overall, in most cases, the global-aware inference could produce summaries of higher quality in a more concise way.

### E.4   Are the Outputs Different from Beam Search from the Beginning?

One may question whether these hypotheses are just the subsets or extensions of beam search hypotheses. The answer lies in the selection of $\beta$. Specifically, the global-aware with a greater $\beta$ (such as $\beta = 12$ for CNN/DM) generates quite different summaries. We plot the count distribution of discrepancy positions between beam search and global-aware inference in Figure 9. It is interesting to find that about one-third of CNN/DM test samples (total count: $11,490$) start to differ at the first word, and most start to differ at the first 5 words. In other words, the global-aware inference generates texts in a way different from beam search that leads to higher scores.

### E.5   Newly Generated Words

Table 10 shows that summaries generated by our method appear more creative. Generally, a greater proportion of novel words implies that the generated text resembles more human-writing, at the same time increases the risk of deviating from reference.

### E.6   Inference Speed

As presented in Table 11, the speed of the global-aware inference is comparable with beam search.

Table 10: The percentage of words in the summary but not in the source.

| CNN/DM | %Novel |
|---|---|
| *reference* | 14.8 |
| *beam search* | 4.5 |
| *global-aware* | **6.7** |

Table 11: Inference speed

| CNN/DM | Tokens/s |
|---|---|
| *beam search* | **25.6** |
| *global-aware* | 23.2 |
| *repetition penalty* | 19.1 |

### E.7 Our Advantages over Predicting Future Rewards

A previous study [18] guides inference by predicting the future automatic matrix score at each decoding step. There are two main disadvantages of training such predictor, (a) there are too many training samples especially when the target sequence is long, making it hard to train; (b) this method forces us to predict every decoding step, greatly slowing the inference speed. By contrast, one appealing benefit of our approach is its high efficiency, since our method only need predict once but can calibrate beam search at each step, which improves beam search from scratch with very little burden of training and inference.

### E.8 Can Our Method Work Together with Attention Head Masking?

There is no doubt that the answer is yes because the attention head masking [1] still need beam search to generate summaries, and our approach is an alternative to beam search. However, we do not recommend the joint usage of the two due to the following three reasons. First, the roles of the two overlap; second, the loss of context information due to masking will negatively affect the global-aware inference; third, adding attention head masking will impose additional burden on training and hyper-parameter searching.

### E.9 Can Predicted Attention Distribution still Lead to the Same Theoretical Result?

Through explaining that the global attention and the predicted attention have the same value range, we will simply prove that the predicted attention distribution is also applicable to the theory in the paper. Since each attention value in the global attention distribution is a cumulative value of the attention probability $\in [0, 1]$ given by the reference tokens, the lower bound tends to be 0 and the upper bound is $1 *$ the number of reference tokens. When the reference is infinite, the upper bound is $\infty$. Therefore, we only need ensure that the predicted attention value is larger than 0 which is constrained by $\exp$.

## F Case Study

Examples are presented in the following pages. We sample some good cases and some bad cases of global-aware inference by the ROUGE-1 $F_1$ score.

Table 12: Generated summaries on CNN/DM sampled by ROUGE-1 $F_1$ score.

| | |
|---|---|
| *Reference* | The ramp agent fell asleep in the plane's cargo hold . He can no longer work on Alaska Airlines flights . |
| *Beam search* (R-1: 25.05) | "Iḿ inside a plane and I feel like itś up moving in the air," the caller says . "There could be a person in there so weŕe going to come back around," the pilot tells air traffic control . The ramp agent is an employee of Menzies Aviation, a contractor for Alaska Airlines . The airline says the man has been permanently banned from working on planes . |
| *Global-aware* (R-1: 43.29) | A ramp agent fell asleep in the cargo hold of an Alaska Airlines flight . The crew and passengers reported unusual banging from the belly of the Boeing 737 . The pilot radioed air traffic control and said he would make an emergency landing . |
| *Global-aware†* (R-1: 50.81) | A ramp agent falls asleep in the cargo hold of an Alaska Airlines plane . The plane makes an emergency landing in Seattle . The man has been banned from working on Alaska Airlines planes . |
| *Reference* | Three people killed; five wounded in attack on attorney general's office in Balkh province . Staff and civilians have been rescued as gunmen engaged Afghan security forces . |
| *Beam search* (R-1: 38.78) | A group of armed assailants stormed into the attorney general's office in Balkh province, northern Afghanistan . Two police officers and a security guard of the provincial attorney general's office were among the dead . Most staff members and civilians have been rescued, but an exchange of fire between Afghan security forces and the assailants is ongoing . |
| *Global-aware* (R-1: 54.00) | Three people were killed and five others were wounded in an attack on the attorney general's office . An exchange of fire between security forces and the assailants is ongoing ." |
| *Global-aware†* (R-1: 44.00) | Gunmen storm attorney general's office in northern Afghanistan . Three people killed, five others wounded . |
| *Reference* | Six young Minnesotans conspired to sneak into Syria and join ISIS "by any means necessary," prosecutors say . The men, ages 19 to 21, were arrested Sunday . They plotted for 10 months, the U.S. attorney for the District of Minnesota says . |
| *Beam search* (R-1: 47.68) | The group of friends, ages 19 to 21, were arrested Sunday . They plotted for 10 months, U .S . Attorney Andrew M . Luger says . Recruiting for the ISIS terrorist network is a problem in Minnesota's Somali community . Four of the men who were arrested appeared in federal court on Monday but did not enter a plea . |
| *Global-aware* (R-1: 61.97) | Six men, ages 19 to 21, were arrested Sunday . They plotted for 10 months, U .S . attorney says . Recruiting for the ISIS terrorist network is a problem in Minnesota's Somali community . |
| *Global-aware†* (R-1: 38.92) | The six men were arrested Sunday . They are accused of plotting to join ISIS in Syria . Recruiting for ISIS is a problem in Minnesota's Somali community . |
| *Reference* | Raul Reyes: In seeking Latino vote, Marco Rubio his own worst enemy on two key issues: immigration reform, Cuba relations . He says on health care, climate change and other issues, he breaks from Latinos' positions. Polls show they don't favor him . |
| *Beam search* (R-1: 35.71) | Ruben Navarrette: Marco Rubio has been his own worst enemy on immigration and Cuba . Navarrette: Rubio has embraced a typical conservative approach to immigration . He says Rubio's political philosophy will be a tough sell to Hispanics . Navarrette: Unless he can offer new ideas, his climb to the Republinomination will be steep . |
| *Global-aware* (R-1: 36.50) | Ruben Navarrette: Marco Rubio is running for the GOP presidential nomination . Navarrette: Rubio has been his own worst enemy on immigration reform and Cuba relations . He says Rubio has embraced a typical conservative approach to immigration . Navarrette: Rubio's political philosophy will be a tough sell to Hispanics . |
| *Global-aware†* (R-1: 40.11) | Ruben Navarrette: Marco Rubio is running for the GOP presidential nomination . Navarrette: Rubio has been his own worst enemy on immigration reform and Cuba relations . He says Rubio has embraced a typical conservative approach to immigration . |
| *Reference* | Police: Yuhei Takashima, 64, says he had sex with girls as young as 14 in Philippines . Officers seize nearly 150,000 photos that the former principal kept of his activities . |
| *Beam search* (R-1: 52.48) | Yuhei Takashima, 64, says he paid for sex with more than 12,000 women in the Philippines . Police seized 147,600 photos that Takashima took of his activities over the years . The ages of the prostitutes he hired ranged from 14 to over 70, he says . |
| *Global-aware* (R-1: 37.33) | Police say Yuhei Takashima, 64, paid for sex with more than 12,000 women . He took 147,600 photos of his activities over more than a quarter of a century, police say . |
| *Global-aware†* (R-1: 49.28) | Police: Yuhei Takashima, 64, paid for sex with more than 12,000 women in the Philippines . He took 147,600 photos of his activities, police say . |
| *Reference* | Kabul faces uncertain future as NATO presence – and the money that came with it – fades away . Interpreters are out of work, NATO trucks sit idle on roads, restaurants are empty . |
| *Beam search* (R-1: 21.95) | Kabul is a city swollen in size but shrunken in scope, anxiously awaiting what comes next . Vast supply chains once kept 120,000 troops fed and watered . Now the bases are gone, and the trucks that once supplied millions sit still . One road, forever pot-holed in the past decade, is now being covered over by Afghans . |
| *Global-aware* (R-1: 16.12) | Afghanistan's capital, Kabul, has been transformed since NATO arrived in 2001 . The city's population has swelled to five times what it was when the U .S . arrived . But the city has also shrunk in scope, with many waiting to see what comes next . |
| *Global-aware†* (R-1: 20.39) | "Kabul, the capital of Afghanistan, has been transformed by the U .S . troop withdrawal . Roads are lined with the detritus of America's war here . Vast supply chains once kept 120,000 troops fed and watered . |

Table 13: Generated summaries on BillSum sampled by ROUGE-1 $F_1$ score.

| | |
|---|---|
| *Reference* | Directs the Secretary of Health and Human Services to establish a process under which a physician may request, in writing from a carrier, assistance in addressing questionable codes and procedures under the medicare program. Sets forth provisions concerning: (1) policy development regarding evaluation and management guidelines; and (2) medicare overpayments. |
| *Beam search* (R-1: 27.61) | Prohibits the Health Care Financing Administration (HCFA) from implementing any new evaluation and management guidelines under the Medicare program unless HCFA: (1) has provided for an assessment of the proposed guidelines by physicians; (2) has established a plan for improving participation of physicians; (3) has carried out a minimum of four pilot projects in at least four different HCFA regions to test such guidelines; and (4) finds that the objectives will be met in the implementation of such guidelines . Requires each pilot project to: (1) be of sufficient length to allow for preparatory physician and carrier education, analysis, and use and assessment of potential E&M guidelines; and (2) be conducted, throughout the planning and operational stages of the project, in consultation with national and State medical societies . |
| *Global-aware* (R-1: 40.00) | Directs the Secretary of Health and Human Services to establish a process under which a physician may request, in writing from a carrier, assistance in addressing questionable codes and procedures under the Medicare program . Requires the carrier to respond in writing within 30 business days and allows a written statement to be used as proof against a future audit or overpayment under the Medicare program . Requires the Administrator of the Health Care Financing Administration to restore the toll-free telephone hotline so that physicians may call for information and questions about the Medicare program .Prohibits the Secretary from implementing any new evaluation and management guidelines under the Medicare program, unless the Health Care Financing Administration: (1) has provided for an assessment of the proposed guidelines by physicians; (2) has established a plan for improving participation of physicians; (3) . . . |
| *Reference* | Reduce Expenditures in Nuclear Infrastructure Now Act or the REIN-IN Act - Prohibits the obligation or expenditure of funds authorized to be appropriated to the Department of Defense (DOD) for FY2014-FY2023: (1) for the research, development, test, and evaluation (RDT&E) or procurement of a long-range penetrating bomber aircraft; (2) to procure an SSBN-X submarine (and prohibits the use of such funds for FY2024 and thereafter to procure more than eight such submarines); or (3) for the RDT&E or procurement of a new intercontinental ballistic missile (ICBM). Prohibits the obligation or expenditure of funds authorized to be appropriated for FY2014 or thereafter for DOD or the Department of Energy (DOE): (1) to make the F-35 Joint Strike Fighter aircraft capable of carrying nuclear weapons; (2) until the Secretary of Defense and the Secretary of Energy jointly certify that the total cost of the B61 life extension program has been reduced to not more than $5 billion; (3) for the W78 life extension program; (4) for the mixed oxide fuel fabrication facility project; (5) to replace the chemistry and metallurgy research building at Los Alamos National Laboratory, Los Alamos, New Mexico; or (6) for the uranium processing facility at the Y-12 National Security Complex, Oak Ridge, Tennessee. Prohibits Navy forces, beginning in FY2020, from including more than eight operational ballistic-missile submarines available for deployment . Prohibits the . . . |
| *Beam search* (R-1: 56.48) | Reduce Expenditures in Nuclear Infrastructure Now Act or the REIN-IN Act Prohibits using funds appropriated to the Department of Defense (DOD) for FY2014-FY2023: (1) for the research, development, test, and evaluation or procurement of a long-range penetrating bomber aircraft; (2) to make the F-35 Joint Strike Fighter aircraft capable of carrying nuclear weapons; (3) until the Secretary of Defense and the Secretary of Energy jointly certify that the total cost of the B61 life extension program has been reduced to not more than $5 billion; (4) for the W78 life extension program; (5) for the reduction of nuclear-armed submarines, beginning in FY2020; or (6) for the National Nuclear Security Administration for FY2024 . |
| *Global-aware* (R-1: 67.03) | Reduce Expenditures in Nuclear Infrastructure Now Act or the REIN-IN Act Prohibits the obligation or expenditure of funds authorized to be appropriated to the Department of Defense (DOD) for FY2014-FY2023: (1) for the research, development, test, and evaluation or procurement of a long-range penetrating bomber aircraft; (2) to make the F-35 Joint Strike Fighter aircraft capable of carrying nuclear weapons; (3) until the Secretary of Defense and the Secretary of Energy jointly certify that the total cost of the B61 life extension program has been reduced to not more than $5 billion; (4) for the W78 life extension program; (5) for the reduction of Nuclear-Armed Submarines, beginning in FY2020; (6) for the SSBN-X submarines; or (7) for the mixed oxide fuel fabrication facility project . Prohibits the obligation or expenditure of funds authorized to be appropriated for FY2014-FY2023 for DOD: (1) to maintain more than 200 intercontinental ballistic missiles (ICBMs), (2) to maintain more than 250 submarine-launched ballistic missiles, (3) for the research, development, test, and evaluation or procurement of a new ICBM, or (4) for the uranium processing facility at the Y-12 National Security Complex, Oak Ridge, Tennessee . Prohibits the . . . |
| *Reference* | Medicare Part D Drug Class Protection Act of 2007 - Amends part D (Voluntary Prescription Drug Benefit Program) of title XVIII (Medicare) of the Social Security Act to require that Medicare prescription drug plans using formularies cover all drugs included in six specified therapeutic categories. Sets forth special requirements for reconsideration of coverage determinations, and appeals for drugs included in such categories. Establishes reporting requirements for drugs in these categories. |
| *Beam search* (R-1: 73.59) | Amends part D (Voluntary Prescription Drug Benefit Program) of title XVIII (Medicare) of the Social Security Act to require prescription drug formularies to cover all drugs in six specified therapeutic categories and classes . |
| *Global-aware* (R-1: 68.45) | Amends part D (Voluntary Prescription Drug Benefit Program) of title XVIII (Medicare) of the Social Security Act to require that the prescription drug formulary include, subject to specified requirements, all or substantially all drugs in each of six specified therapeutic categories of covered Medicare part D drugs: (1) Immunosupessants; (2) Antidepressants; (3) Anticonvulsants; and (4) Antiretrovias . Provides for special coverage of drugs included in specified therapeutic categories during determinations, reconsiderations, and appeals . |

Table 14: Generated summaries on arXiv sampled by ROUGE-1 $F_1$ score.

| Reference | in this paper , we develop a structure - preserving discretization of the lagrangian framework for electromagnetism , combining techniques from variational integrators and discrete differential forms . this leads to a general family of variational , multisymplectic numerical methods for solving maxwell s equations that automatically preserve key symmetries and invariants . in doing so , we demonstrate several new results , which apply both to some well - established numerical methods and to new methods introduced here . first , we show that yee s finite - difference time - domain ( fdtd ) scheme , along with a number of related methods , are multisymplectic and derive from a discrete lagrangian variational principle . second , we generalize the yee scheme to unstructured meshes , not just in space but in @xmath0-dimensional spacetime . this relaxes the need to take uniform time steps , or even to have a preferred time coordinate at all . finally , as an example of the type of methods that can be developed within this general framework , we introduce a new asynchronous variational integrator ( avi ) for solving maxwell s equations . these results are illustrated with some prototype simulations that show excellent energy and conservation behavior and lack of spurious modes , even for an irregular mesh with asynchronous time stepping . |
|---|---|
| Beam search (R-1: 37.82) | the yee scheme is one of the most successful numerical methods used in the field of computational electromagnetics , particularly in the area of microwave problems . although it is not a high - order " method , it is still preferred for many applications because it preserves important structural features of maxwell s equations that other methods fail to capture . in this paper , we show that these desirable properties are direct consequences of the variational and discrete differential structure of the yee scheme , which mirrors the geometry of maxwell s equations . moreover , we show how to construct other variational methods that , as a result , share these same numerical properties , while at the same time applying to more general domains . |
| Global-aware (R-1: 45.96) | the yee ( finite - difference time - domain ) scheme is one of the most successful numerical methods used in the field of computational electromagnetics , particularly in the area of microwave problems . in this paper , we show that these desirable properties are direct consequences of the variational and discrete differential structure of the yee scheme , which mirrors the geometry of maxwell s equations . moreover , we show how to construct other variational methods that , as a result , share these same numerical properties , while at the same time applying to more general domains . the key symmetry of maxwell s equations is not rotational or translational symmetry , as in mechanics , but a differential gauge symmetry . without taking additional care to preserve this gauge structure , even variational integrators can not be expected to capture the geometry of maxwell s equations . this differential / gauge structure also turns out to be important for the numerical performance of the method , and is one of the hallmarks of the yee scheme . this differential / gauge structure also turns out to be important for the numerical performance of the method , and is one of the hallmarks of the ye . . . |
| Reference | we present a new general mechanism to generate curvature perturbations after the end of the slow - roll phase of inflation . our model is based on the simple assumption that the potential driving inflation is characterized by an underlying global symmetry which is slightly broken . |
| Beam search (R-1: 36.88) | most inflationary models assume that there is only one scalar field involved in the dynamics of inflation . in this work we point out a qualitatively new effect that might arise if one relaxes the assumption of a single dynamical field . in a multi - field scenario in which the inflationary potential is characterized by a broken symmetry , the quantum fluctuations generated during the inflationary stage represent fluctuations in the initial conditions for the dynamics of the inflaton in the subsequent stage , thus implying that the background dynamics after the slow - roll phase has ended will differ in different regions of the universe . since the background fields are coupled to the other fields into which they decay , the fluctuations generated during the slow - roll phase will affect the subsequent decay process . the present work , assuming that the inflaton decay into other fields through the non - perturbative process of preheating , is then aimed to understand whether isocurvature inflaton fluctuations , generated during the slow - roll stage , can lead to perturbations of the background metric through variations of the preheating efficiency . while the generation of curvature perturbations during the stages following the slow - roll phase has already been considered in some works , the . . . |
| Global-aware (R-1: 53.46) | we show that in a multi - field scenario in which the inflationary potential is characterized by a broken symmetry , quantum fluctuations generated during the slow - roll stage of inflation may affect the efficiency of the subsequent preheating process . we obtain a general formula for the curvature perturbations generated from an inhomogeneous preheating efficiency related to the quantum fluctuations produced during inflation . the present work is then applied to the case of a broken @xmath0 symmetry . |
| Reference | in this work we investigate an influence of decoherence effects on quantum states generated as a result of the cross - kerr nonlinear interaction between two modes . for markovian losses ( both photon loss and dephasing ) , a region of parameters when losses still do not lead to destruction of non - classicality is identified . we emphasize the difference in impact of losses in the process of state generation as opposed to those occurring in propagation channel . we show moreover , that correlated losses in modern realizations of schemes of large cross - kerr nonlinearity might lead to enhancement of non - classicality . |
| Beam search (R-1: 47.48) | decoherence due to losses is a main obstacle to implementations of schemes using kerr and cross - kerr nonlinearities . in our work we discuss an aspect of the decoherence which has been seldom discussed when considering an influence of losses on states generated via kerr nonlinearity . namely , we address losses arising in the process of generation and not due to propagation of the generated state via lossy channels . we concentrate our attention on a feature that might be quite significantly pronounced in modern schemes of generating large kerr nonlinearity : the modal loss can be strongly correlated . indeed , the modes occupy the same volume and interact with the same physical systems which form the reservoirs . also , if the kerr - nonlinearity scheme implies a sufficiently strong dispersive coupling of light modes to emitters , then coupling of these emitters to dissipative reservoirs might also appear to be quite strong . as a result , this would mean strongly correlated modal losses . on a number of examples we show how the correlated loss enhances and creates intermodal correlations and even entanglement , and might lead to generation of entangled states quite . . . |
| Global-aware (R-1: 41.50) | decoherence of quantum states generated via kerr and cross - kerr nonlinearities is discussed . it is shown that modal loss can be strongly correlated , i .e . , it can be decoupled from uncorrelated reservoirs . this strongly correlated loss enhances and creates intermodal correlations and even entanglement , and might lead to generation of entangled states quite different from those generated in the same scheme without loss . the strongly correlated loss can result in the significantly enhanced robustness of the generation scheme . |

Table 15: Generated summaries on Multi-News sampled by ROUGE-1 $F_1$ score.

| | |
|---|---|
| *Reference* | – Joy Milne noticed a small difference in her husband just before he turned 40 . "His smell changed," the Scottish woman tells the BBC . "It wasnt́ all of a sudden . It was very subtle—a musky smell ." He was diagnosed with Parkinsoń disease six years later and died in June at age 65 . Milne found herself supporting the Parkinsoń UK charity, where she noticed others had the same distinct smell . She happened to mention her observation to scientists at a talk, and they later put her to the test, presenting her with T-shirts worn by six people with Parkinsoń and six without . "She got the six Parkinsoń, but then she was adamant one of the ćontrolśubjects had Parkinsoń," says a scientist . Researchers were impressed and dubbed her a "super-smeller," per AFP . Then eight months later, that one control subject returned to tell scientists he had just been diagnosed with Parkinsoń . Because of Milneś sniffer, scientists now theorize . . . |
| *Beam search* (R-1: 48.93) | – Joy Milneś husband, Les, was diagnosed with Parkinsoń disease at age 45—six years before he died in June . "His smell changed and it seemed difficult to describe . It wasnt́ all of a sudden . It was very subtle—a musky smell," the Australian woman, who lives in Perth, tells the BBC . "I got an occasional smell ." After joining the charity Parkinsoń UK, she met people with the same odor, and researchers at Edinburgh University decided to test Milneś ability to detect the disease using t-shirts worn by six people with Parkinsoń and six without . "Her job was to tell us who had Parkinsoń and who didnt́," says Dr . Tilo Kunath . "Her accuracy was 11 out of 12 . We were quite impressed ." Six of the Parkinsoń patients wore the shirts for a day, while the other six wore them for a week . Milne correctly identified six of the Parkinsoń patients and six of the non- Parkinsoń patients, but she was "adamant" that one of the non- Parkinsoń subjects had the disease, Kunath says . Eight months later, he told Kunath hed́ been diagnosed with Parkinsoń . . . |
| *Global-aware* (R-1: 55.18) | – When Joy Milneś husband was diagnosed with Parkinsoń disease six years before his death in June, her sense of smell told her something was amiss . "His smell changed and it seemed difficult to describe . It wasnt́ all of a sudden . It was very subtle—a musky smell," she tells the BBC . After joining the charity Parkinsoń UK, she met people with the same odor . She mentioned it to scientists at the University of Edinburgh, and they decided to test her—and were "quite impressed," says one of the scientists . Milne correctly identified six people with Parkinsoń and six without it using T-shirts that had been worn by six people with Parkinsoń and six without it . Her accuracy was 11 out of 12 . "She got the . . . |
| *Reference* | – President Obama devoted his weekly radio address today to the Connecticut school shooting, asking Americans to pray for the victims and their families . "Most of those who died were just young children with their whole lives ahead of them," he said, echoing yesterdayś news conference . "And every parent in America has a heart heavy with hurt ." Obama reiterated that itś time to "come together to take meaningful action" to prevent future shootings, thought he stopped short of specifics, reports Reuters . (USA Today has the full transcript here .) Advocates of stricter gun control already have begun to step up pressure, including New York City Mayor Michael Bloomberg . "We have heard all the rhetoric before," he said yesterday . "What we have not seen is leadership—not from the White House and not from Congress . That must end today ." |
| *Beam search* (R-1: 59.23) | – "Most of those who died were just young children with their whole lives ahead of them . And every parent in America has a heart heavy with hurt," President Obama said today in his weekly radio address, reports USA Today . "There are families in Connecticut who cant́ do that today . And they need all of us now . All of us can extend a hand to those in need—to remind them that we are there for them, that we are praying for them ." Obama again stopped short of calling for tougher gun laws, reports Reuters . "We grieve for the families of those we lost," he said . "And we keep in our prayers the parents of those who survived ." |
| *Global-aware* (R-1: 48.81) | – President Obama today repeated his call for "meaningful action" to reduce gun violence in the wake of the Connecticut school shooting, reports Reuters . "We grieve for the families of those we lost," he said in his weekly radio address . "And we keep in our prayers the parents of those who survived ." Obama didnt́ call for stricter gun laws, notes USA Today, but he did urge a national conversation . "This weekend, Michelle and I are doing what I know every parent is doing—holding our children as close as we can and reminding them how much we love them," he said . "There are families in Connecticut who cant́ do that today . And they need all of us now ." |
| *Reference* | – Jesse Jackson Jr . wept in court today, apologized for using $750,000 in campaign funds as his personal piggy bank, and then took his lumps: 30 months in federal prison and three yearsṕrobation, reports the Chicago Tribune . "I am the example for the whole Congress," he said . "I understand that . I didn't separate my personal life from my political activities, and I couldn't have been more wrong ." The former Illinois congressman added an unusual request: He wants to serve his time in Alabama because "I want to make it a little inconvenient for everybody to get to me ." His wife, Sandi, then got sentenced to 12 months in prison for filing false tax returns about their income, reports AP . She wept, too . "My heart breaks every day with the pain this has caused my babies," she said, referring to their two children, ages 9 and 13 . Because of the kids, the judge will . . . |
| *Beam search* (R-1: 52.74) | – Former congressman Jesse Jackson Jr . and his wife, former Chicago alderman Sandi Jackson, were sentenced to prison today, reports the Chicago Sun-Times . Jesse Jackson Jr . got 2 1/2 years for using campaign funds for personal expenses, while his wife got one year for filing false tax returns . The Jacksons had pleaded for mercy for each other in court . Jesse Jackson Jr . "I didn't separate my personal life from my political life, and I couldn't be more wrong," he said . "I take responsibility for my actions ." Sandi Jackson said she needed to be with her children . "I ask to continue to provide for my children," she said . "To take the mother away . . . would be an unbearable burden on these two children ." |
| *Global-aware* (R-1: 47.29) | – Former congressman Jesse Jackson Jr . and his wife, former Chicago alderman Sandi Jackson, were sentenced to prison today, reports the Chicago Sun-Times . Jackson was sentenced to 2 1/2 years for using campaign funds for personal expenses, while his wife got a year for filing false tax returns . The Jacksons had pleaded for mercy for each other in court, but the judge didnt́ seem swayed, reports AP . "There may be blurred lines for congressmen to follow when their lives are political . This case did not come near those areas," said Judge Amy Berman Jackson . "I cannot do it and I will not do it ." She rejected Jackson Jr .ś defense that his bipolar disorder played a role, saying his string of accomplishments—" propped up by a political family dynasty"—"points to only one conclusion, and that is that you knew better ." She also rejected . . . |

# G  Global-aware Inference in Neural Machine Translation

Though the focus of this paper is Neural Abstractive Summarization, we will share how to enable global-aware inference to improve beam search significantly in NMT using a simple variation.

The gap between NMT and NAS is that NAS is a more-to-less generation task where the global attention values of many words are very low, which makes it relatively easy to trigger punishment at the beginning. As one can see in App. E.4 and the examples provided, many global-aware summaries are different from that of beam search from scratch, implying that the global scoring mechanism is activated in the very early stage. Since translation is a nearly one-to-one task, it is more likely to ignore some locality biases during the beginning stage when the local attention is generally low. Notably, although there is no very low global attention in translation tasks, their global attention distributions are still irrelevant with uniform distributions (here we only consider pre-trained translation models).

Table 16: BELU results of WMT16. We perform the blocked global-aware inference with the block length of 10.

| $\mathcal{K} = 5$ | BELU |
|---|---|
| **beam search** | 37.6 |
| **ORACLE global-aware** | 39.2 |
| **+ Blocking** | **41.6** |
| **Global-aware** | 37.8 |
| **+ Blocking** | **38.1** |

To alleviate the gap, a straightforward-but-effective approach is to transform the one-to-one generation task to several more-to-less tasks. We divide the reference into several blocks of equal length and predict the global attention distribution of each block. Taking a reference of 25 tokens as an example, if we set the block length as 10, then there will be three blocks, namely the block of $1 - 10$ tokens, $11 - 20$ tokens, and the last 5 tokens. We use the special token $b_1$ to denote the first block, $b_2$ to denote the segment composed of the first and second block, and $b_0$ to denote the whole reference. When we train the attention-prediction model, these special tokens are concatenated before the source document to tell the predictor which segment it should predict. During inference, we first predict the global attention distribution of $b_0$ and calculate the optimal length, e.g. 32, then we can figure out there are three more global attention distributions that need be predicted, i.e., $b_1$, $b_2$ and $b_3$. We apply the predicted attention distribution of $b_1$ to guide the inference at the beginning, and the attention distribution is transformed to that of $b_2$ when the length of generated sequence is larger than 10 and so on. When the decoding step reaches 30, the attention distribution of $b_0$ is deployed until all candidate sequences are terminated. Although this blocking operation would increase the difficulty of training and inference, these negative effects are controllable by adjusting the block length.

We use mBART [23] as the neural translation model to examine the blocking operation on WMT16 en-ro sentence-level translation dataset, and related results are presented in Table 16. Compared with beam search, our proposed global-aware elevates the BELU score from 37.6 to 37.8, and ORACLE global-aware improves the result to 39.2. The blocking operation further improves global-aware to 38.1, and ORACLE global-aware to 41.6. We also experiment this operation on summarization tasks where minor improvement is observed.

Finally, since the blocking operation would inevitably increase the difficulty of training and inference and is not conducive to the robustness, it may not be the most reasonable direction for improving the global-aware inference in one-to-one tasks. We think there are three following issues that might be worth exploring in the future. 1) Whether the proposed global-aware inference has already been able to improve document-level translation significantly; 2) intuitively, we averaged the global attention distributions of all layers, but it is possible that the global attention in a single layer or a combination of several layers can better guide decoding; 3) whether a more comprehensive global protocol can be found to regulate beam search in a more detailed way.