# OpenReview forum: "Global-aware Beam Search for Neural Abstractive Summarization"
_NeurIPS.cc/2021/Conference — NeurIPS 2021 Poster_

### Official Review · Reviewer_M9Ny · 2021-07-16

**Rating:** 6
**Confidence:** 2

**Summary:**

This paper thinks that cross attention distribution in transformer’s encoder-decoder framework can help the performance of beam search. At the same time, this paper also proposes an algorithm to predict distribution to help beam search. This paper conduct experiments on 9 datasets to prove its effectiveness.


**Ethics Review Area:**

["I don’t know"]

**Limitations And Societal Impact:**

see review

**Main Review:**

There is no doubt that the empirical results given in this paper are quite strong, but it is difficult for me to understand every detail of the paper and the motivation of the paper.

Is there a possibility that the performance improvement is obtained by the increase of model parameters for attention prediction? Usually, better performance can be obtained by increasing the model parameters.
"""
197 Implementation Details. We adopt a randomly initialized 2-layer transformer-encoder in the
198 attention-prediction model, and the structure of each layer is the same as the BART-encoder layer.
"""


**Time Spent Reviewing:**

6

---

> ### Author Response · Authors · 2021-08-10
> **Response to  Reviewer M9Ny**
>
> Thank you for your positive comments on our paper!
>
> In this paper,  we did not make any modifications on BART and PEGASUS parameters, so we actually used the same generator as beam search to generate summaries. The only difference is that we additionally introduce a predicted global attention distribution to guide beam search. Therefore, the interesting point is that we generate totally different summaries from beam search using the same text generator. In this era of pre-training, it usually requires a stronger pre-training model to obtain significant and stable improvement on multiple data sets, but the cost is huge. In contrast, we only spend little effort to train a small model, but achieve similar improvement results.

---

### Official Review · Reviewer_9iis · 2021-07-16

**Rating:** 6
**Confidence:** 4

**Summary:**

This paper proposes a generalization of the attention coverage penalty in seq2seq models (Wu et al 2016), where instead of encouraging a uniform coverage of the source, this work first predicts the expected coverage of each source word. During beam search, beam hypotheses that violate the expected attention coverage constraints receive a penalty. Experiments on 9 summarization benchmarks show that the proposed coverage loss outperforms using existing attention coverage losses during beam search.

**Limitations And Societal Impact:**

Yes

**Main Review:**

Strengths:
1. This work generalizes the attention penalty loss, where different variants are used in different applications (for example, translation often encourages uniform coverage of the source, whereas summarization often penalizes repeatedly attending to the same word to reduce repetitions). In theory, the expected attention coverage predictor can learn to vary its behavior according to the task, instead of relying on the expert design of the form of coverage loss.
2. The approach proposed by this paper can be viewed as maintaining similar statistics between training and testing (similar to professor forcing), and this might inspire future research matching other statistics such as the distribution of hidden states.
3. Experiments show improvements over existing coverage mechanisms.

Weaknesses:
1. In Table 4, the last row shows that Global Aware has R1, R2, RL of 45.57, 22.60, 37.61, but according to table 3, this is the performance of Global Aware without length reward, whereas the performance of Global Aware in table 3 is 45.44, 22.15, 37.11.
2. While the proposed approach might alleviate the locality bias of beam search, it still suffers from a similar locality bias. The attention constraints (Eq 7) only become active when the accumulated attention exceeds its total budget, which I expect to happen rarely at the beginning of the search (when accumulated attentions are small) and more frequently towards the end of the search (when accumulated attentions are larger). Therefore, I wouldn't expect this "global term" to guide early decisions during search, and this is exactly where beam search has lots of uncertainty and global planning is needed the most. For this reason, I don't think the proposed approach (or existing attention coverage penalties) is truly "global aware" unless there's some estimate of future rewards (https://arxiv.org/pdf/1701.06549.pdf).
3. In practice, the proposed approach introduces additional components: an expected total attention predictor, as well as additional hyperparameters. While using empirical hyperparameters for some settings results in good performances, I'm not totally convinced how robust the performance is with respect to hyperparameters, and how much tuning effort is needed.
4. The comparison to the baseline coverage in table 4 does not appear fair. According to my understanding coverage doesn't include length penalty, and if we compare baselines in table 4 with Global Aware without length reward in table 3, coverage underperforms on XSUM but outperforms on CNN/DM, and the other baselines also appear to have similar performances to the proposed method.
5. In Figure 2, the performance changes are not significant, and I think using even larger beam sizes would show a clearer trend.

Questions for authors:
1. How well does your expected attention predictor predict attentions? Is it predictable just from the source? Wouldn't different summaries for the same document pick source words differently?
2. In Eq 8 why not directly use the final score at the current step G(b_{m-1}^t) but take the product of all steps observed so far? If the constraints are satisfied at step t, doesn't that mean it's always satisfied at steps <t?


Presentation Issues:
1. The length reward function R (Eq 10) should be in the main paper.
2. Section 2, "Note that both models used in this paper are fine-tuned and parameter-fixed." what does this mean? Is it fixed or finetuned?
3. Line 150, "its global score at the termination is the maximum 1," Do you mean g_i?

Suggestions:
1. While the appendix shows some decoding examples, I think it would be helpful to visualize how this additional loss affects the search process by visualizing which constraints are "active" and which beam hypotheses are dropped due to incurring penalties on those constraints.

Overall, this is an interesting work generalizing widely-used attention coverage losses. However, the approach taken in this paper needs to add a predictor and I'm not convinced how robust the performance is with respect to extra hyperparameters. Besides, I have some doubts about some results. Therefore, I'm not recommending its acceptance in its current form.

==== Post Rebuttal ====

Overall, some of my concerns have been addressed, but there seems to be some misunderstanding about previous works (it's equally possible that I might have misunderstood how attention coverage works). Therefore, I'm keeping my score for now and would appreciate it if the authors could post a short clarification.

==== Post Post Rebuttal ====

After discussing with the authors, I found that while bottom-up summarization uses stepwise coverage penalty, GNMT might not have used stepwise coverage. Therefore, my concern about wrong baseline implementation was not well founded, and I'm increasing my score to 6.

**Time Spent Reviewing:**

3

---

> ### Author Response · Authors · 2021-08-10
> **Response to Reviewer 9iis, Part 1**
>
> Thank you for your valuable comments.
>
> First of all, although coverage penalty has inspired us, we want to emphasize that we don't think this method is a simple generalization of coverage penalty due to completely different motivations and stepwise calibration mechanism. Specifically, coverage function desires the total attention of each source token allocated by the hypothesis should surpass a preset threshold, so as to encourage the hypothesis to cover more source information, but is only activated at the termination. By contrast, global-aware desires that the growing attention of each source token (i.e., local attention) should not surpass the global optimal attention value during decoding and exactly reach the optimal attention value at the termination. More comparison details have been presented in Appendix E.
>
>
> ### [W1. About Table 3-4]
> Sorry for the confusion. In XSUM of Table 3,  the global-aware result is given when $\gamma=1$ (in order to show the ablation study). It proves that length reward is less effective in XSUM though it could boost the performance on CNN/DM. Table 4 presents the best results obtained by hyper-parameter searching, i.e.  $\gamma=0$ or the length reward was not applied. Related experiment setup has been reported at Line 212.
>
> Less effectiveness of length reward could be attributed to the fact that XSUM has relatively short length. It is noted that we have actually discussed this in Line 228--231 of the submission. Further discussion can be seen in the response to W4 shortly.
>
> ### [W2. About the locality bias]
> We agree that the proposed approach may have some locality bias at the beginning stage. However, this problem has little impact on summarization, which is the focus of our paper. Our experimental results actually indicated that the "global term" takes effect in the very early stage.
>
> 1) Theoretically, since summarization is a more-to-less generation task, the global attention of most words is very low so that it is relatively easy to trigger punishment even if the fault appears at the beginning.
>
> 2) Empirically, we did a simple experiment in Appendix D.4 to show that the global scoring function is activated in the beginning stage, where our outputs start to deviate from that of beam search. By Figure 5, we can see nearly half of hypotheses become different from that of beam search from the first 5 words. You can also browse our examples to see more difference cases between global-aware and beam search.
>
> 3) Importantly, one major appealing benefit of our approach is its high efficiency, since our method only needs to predict once but can calibrate beam search at each step, which improves beam search from scratch with very little burden of training and inference. This presents a huge advantage compared with the arXiv paper mentioned by you, where a predictor must be trained to predict the automatic matrix score for each decoding step. To highlight the main distinction, we will conduct a detailed discussion with the arXiv paper in the appendix.
>
> 4) Indeed, this problem may have impact on one-to-one generation tasks like translation. To examine this, we  have actually designed a model variant  to alleviate the problem by transforming a one-to-one task into several more-to-less tasks. Instead of predicting the optimal attention distribution at each decoding step, to preserve its efficiency as much as possible，we came up with a compromise approach, i.e., dividing the reference into several blocks of equal length and predicting the global attention distribution of each block. Then, the local attention in one block is desired not to surpass the global attention of that block. The experiment shows the blocking operation could improve global-aware on NMT (about 0.3 BELU score), but fails on NAS, also implying that the proposed global-aware is active from the beginning stage in NAS. This variant was originally used as our follow-up work, but now that you mention this problem, we will present its details in the appendix of this paper.
>
>
> ### [W3. Additional components and additional parameters?]
> 1)We do introduce a component of the expected total attention predictor, the number of hyper-parameters is actually **LESS** than that of beam search and **LESS** effort is needed in our hyper-parameter searching than in that of beam search.
>
>  - Though we introduce two new hyper-parameters $\beta$ and $\gamma$, we discard three hyper-parameters commonly-used in beam search, namely length normalization $a$, min length and max length (see line 208--210).
>
>  - Less effort is needed in hyper-parameter searching than beam search, since the length regularizations of beam search are very sensitive to sentences of different lengths. By comparison, since the global-aware inference predicts sample-wise optimal length, we only need to choose a rough $\gamma$ to balance the relationship between length reward and global score. As we mentioned in line 213, we recommend $\gamma=1$ for most cases (8 of 9 datasets adopt the setup), or a more cautious way is to try both $\gamma=0$ and $\gamma=1$ if references of a dataset are all very short.
>
> 2)Additional sensitivity analysis in the test set shows the robustness of our method versus these hyper-parameters.
>
> In CNNDM, when $\gamma=1$,
>
> | $\beta$ | 2 | 4 | 6 | 10|  12 | 15 | 18 | 20 |
> |  ----  | ----  |  ----  | ----  |  ----  | ----  |  ----  | ----  | ----  |
> | R-L   | 41.32 | 41.49 | 41.68 | 41.95 | 42.01 | 42.04 | 42.03 | 42.00 |
>
> When $\beta=12$,
>
> | $\gamma$ | 0 |0.5|1|1.5|2|
> | ----  |  ----  | ----  |  ----  | ----  | ----  |
> |R-L  |41.41| 41.92| 42.01| 42.04| 42.04|
>
> We can see the performance is less sensitive to the hyper-parameters when $\beta \geq 10$ or $\gamma \geq 1$.
>
> Then, in XSUM, when $\gamma=0$,
>
> | $\beta$ | 2 |4|6|10|12|15|18|20|
> |  ----  | ----  |  ----  | ----  |  ----  | ----  |  ----  | ----  | ----  |
> |R-L  |37.56| 37.61| 37.63| 37.65| 37.58| 37.56| 37.55| 37.52|
>
> When $\beta=4$,
>
>  |$\gamma$ | 0 |0.5|1|1.5|2|
> | ----  |  ----  | ----  |  ----  | ----  | ----  |
> |R-L|37.61 | 37.29| 37.11| 37.07| 37.05|
>
> There is a significant decline when applying length reward in XSUM. In fact, global score favors shorter hypotheses (line 149--153), and length reward could alleviate the bias. However, If the references of a data set are already very short such as XSUM,  length reward may lead to a counterproductive effect.
>
> ### [W4: about the coverage mechanism in Table 4]
> Thanks for your comments but we disagree with your opinion that coverage doesn't include length penalty, respectfully.
>
> 1) In general, the specific operation of coverage penalty in this paper is implemented as follows.  We first use beam search whose hyper-parameters of length penalty include length normalization, the minimum, and the maximum length to generate $K$ candidate hypotheses, where hyper-parameters follow the BART paper. Next, we use our predicted global attention distribution to replace the preset threshold in the original paper to select the most reasonable hypothesis (see line 234--236). Therefore, length penalty has actually been considered in Coverage already.
>
> 2) To fairly examine the effect of the global aware method, we also conduct an additional experiment of Coverage by replacing its original length penalty with the length reward.
> We obtain the new results: R-1 44.79, R-2 21.68, R-L 41.55, which are still significantly lower than global-aware (45.13, 21.77, 42.04).
>
> 3) Actually, Coverage could not improve beam search fundamentally because it only works at the termination step, which means the final hypothesis it generates is often similar to that of beam search, or a subset or an extension of beam search hypothesis. We also calculate the ORACLE results of the Coverage mechanism and obtain 46.18, 23.10, 42.92 on CNN/DM, which are even much lower than ORACLE global-aware without length reward (50.46, 27.53, 47.43).
>
>
> ### [W5: about Figure 2]
> We will follow your suggestion to draw the trend plot with some larger beam sizes.

---

> ### Author Response · Authors · 2021-08-10
> **Response to Reviewer 9iis, Part 2**
>
> ### [Q1: about the predictor]
>
> - [a. How well does your expected attention predictor predict attentions?]  Frankly, we didn't put much effort into the prediction model because that's not the focus of our paper. If more efforts are made in model design and parameter adjustment, we believe it will be better.
>
> - [b. Is it predictable just from the source?] Essentially, global attention contains the model knowledge because it is calculated by the trained model parameters, and these knowledge is regular and traceable. This knowledge tells the decoder how to allocate attention, not only from the perspective of token importance, but also from the features that attention model can capture, such as syntax and word order. Although the model knowledge will be reflected differently in different tasks or data sets, it does not affect our prediction from source because similar sources should have similar attention distribution in most cases, just as they have similar reference. In a word, attention distribution is a continuous form of the reference obtained by a series of regular transformations, allowing the reference to be predicted as a whole rather than word-by-word.
> - [c. Wouldn't different summaries for the same document pick source words differently?] At present, the data sets we use are one source corresponding to one reference, but in the future, we may study the task of one source with multiple references.
>
>
> ### [Q2: about Eq.8]
> The main reason is to balance the proportion between global score and probability. The final probability score is a cumulative value, which means that at each step there is a new $\log p$ added into the probability score. Accordingly, the final global score should also be a cumulative value, and there should  also be a $\log \mathcal{G}$ added into the global score at each step ($p$ and $\mathcal{G}$ have the same range, i.e., [0,1]). If we don't use the cumulative global score, then as the inference proceeds, the influence of probability score on the joint score will be greater and greater, which is obviously not what we want to see.
>
> ### [Other response]
> - [a. issue 2] We download the fine-tuned parameters of BART and PEGASUS from HuggingFace (https://huggingface.co/) and fix them. In this paper, we didn't do any modifications to these parameters.
>
> - [b. issue 3] It is not $g_i$. This sentence means the global score $\mathcal{G(\cdot)}$ of a desired hypothesis should be equal to 1 at the termination step.
>
>
> ### [Discussions on why we need a predictor]
> It is well-known that there is an exposure bias with teacher forcing training, but we can't deny that it is a very efficient training method. Its high efficiency and ease of use make it possible to train expensive large models. Therefore, to alleviate exposure bias while preserving these hard-won pre-training parameters achieved by teacher forcing training, an additional model distribution is indeed necessary to guide inference (line 279--284 expresses the similar meaning). This additional model should not suffer from exposure bias, just like our attention-prediction model.
>
> Finally, thank you again for sharing your thoughts on our paper. We really enjoy the process of response with you which we believe could enhance our paper further.

---

> > ### Comment · Reviewer_9iis · 2021-09-01
> > **how does existing attention coverage work?**
> >
> > Thanks for the detailed response!
> >
> > First, I want to point out that the row [2] (bottom-up abstractive summarization) of Table 4 in Appendix E is wrong --- in my understanding, the coverage penalty is applied stepwise (see https://github.com/sebastianGehrmann/bottom-up-summary#f-use-probabilities-in-bottom-up-attention, in python translate.py there's a flag --stepwise_penalty, which will invoke a function call https://github.com/sebastianGehrmann/OpenNMT-py/blob/copy_constraint/onmt/translate/Beam.py#L91, which will further call https://github.com/sebastianGehrmann/OpenNMT-py/blob/copy_constraint/onmt/translate/Beam.py#L235 per each step during beam search). Based on this observation, I think it is fair to say that this paper can be viewed as a generalization of existing attention coverage mechanisms: the correct "Desired Situation" for [2] should be $\sum_{m=1}^t \alpha_{t,i}\le 1$, whereas the proposed approach (the last row) is $\sum_{m=1}^t \alpha_{t,i} \le g_i$. Therefore, it seems to me that this paper generalizes 1 to $g_i$, and the statement "Specifically, coverage function desires the total attention of each source token allocated by the hypothesis should surpass a preset threshold, so as to encourage the hypothesis to cover more source information, but is only activated at the termination." is not generally true.
> >
> > W1. I see, so it's due to hyperparameter selection. Thanks for the clarification!
> >
> > W2. 1. It makes sense that $g_i$'s are usually small (would be nice to include some visualiations for the learned $g_i$'s as well). 2. But I don't think Figure 2 in Appendix D.4 fully addresses my concern. In my understanding Figure 2 is looking at the final hypothesis, so the future scores might affect which earlier words are eventually selected, whereas a more direct metric is to measure the overlap of the beam at each step (between your approach and no coverage penalty).
> >
> > W3. 1. I see your point that the proposed method has fewer hyperparameters, but the min length and the max length in beam search have intuitive meanings and are practically easy to determine: max length is usually limited by compute and people usually use default values (e.g., fairseq has a default value of 200), and min length can be set based on simple dataset statistics. Instead, the hyperparameters here need to measure on validation set. Besides, the attention predictor needs to trained as well which adds further parameters.
> >
> > "we recommend $\gamma=1$ for most cases (8 of 9 datasets adopt the setup)": according to L212-213, $\gamma=1.5$ for CNNDM and $\gamma=0$ for XSUM, so it seems that at least two datasets did not adopt the recommended setup?
> >
> > 2. Thanks for the sensitivity analysis! It seems that the performance is fairly stable, although I still doubt if it is practical to tune for $\gamma\in \\{0, 1\\}$ and $\beta \in \\{2, 4, 6, 10, 12, 15, 18, 20\\}$.
> >
> > W4. 1. Thanks for the clarification that length penalty is already included by default.
> >
> > 3. I don't understand why you keep stating that coverage only works at the termination step: both Gehrmann et al "bottom-up abstrative summarization" and See et al "get to the point" use stepwise coverage penalty, and to the best of my knowledge, Google's NMT (GNMT) also uses stepwise coverage penalty (see https://github.com/tensorflow/addons/blob/v0.13.0/tensorflow_addons/seq2seq/beam_search_decoder.py#L1016). In your implementation of baselines, is attention coverage penalty only applied to the finalized hypotheses? If so, isn't that only reranking the final K hypotheses and make no difference during beam search? This would be crucial to clarify.
> >
> > Q1. My point is that since each source might correspond to multiple valid targets (that's different from how many targets are collected in the dataset), e.g., there might be multiple valid ways of summarizing an article, and if different targets correspond to different source coverage patterns (e.g., in the extreme case, one person might summarize by only picking the first half of the article, and the other person might only pick the second half of the article, while either version is valid, the mix of both is not, then what should the attention predictor do?), that seems to invalidate the basic assumption of this approach that the attention coverage is to some extent predictable from the source alone.
> >
> > Overall, some of my concerns have been addressed, but there seems to be some misunderstanding about previous works (it's equally possible that I might have misunderstood how attention coverage works). Therefore, I'm keeping my score for now and would appreciate it if the authors could post a short clarification.

---

> > > ### Author Response · Authors · 2021-09-01
> > > **Further response**
> > >
> > > Thank you for your additional comments which could further clarify and enhance the paper. We double check the literature again. Interestingly, we find that the penalty functions described in GNMT [1] and bottom-up [2] papers were indeed both activated at the termination only. However, after reading through the released codes, a step-wise modification was actually applied on [2]'s function practically. Nevertheless, We argue that this operation didn't change the motivation of [2] and is still much different from ours.
> > >
> > > 1) [Functions in [1] and [2]  are not dependent on step $t$]. Taking [2] as one example,  we can see the coverage function of [2] is $cp(x;y) = \beta(-n+\sum_{i=1}^n\max(1.0,\sum^m_{j=1} \alpha^j_i)$, where these notations are explained as "Throughout this paper, we consider a set of pairs of texts $(X , Y)$ where $x \in X$ corresponds to source tokens $x_1$, . . . , $x_n$ and $y \in Y$ to a summary $y_1$, . . . , $y_m$ with $m \ll n$.". From its left side, it is very clear that the function cannot evaluate the generated sequences of each step as $y$ is a whole summary. It is also noted that $m$ here is equivalently $T$ in our paper.
> > >
> > > 2) [Their Motivations]. [1] and [2] actually have different motivations. In our opinion, step-wise penalization may be impossible in [1]. Specifically, [1] encourages the final translation to cover the source. However, it is not reasonable to penalize an unfinished translation during decoding just because it fails to cover the source (since this unfinished translation might reach the expectation as the generation proceeds). The coverage motivation we actually talked about in the earlier rebuttal is about [1]. On the other hand,  although in [2] its penalty function was named as the coverage function, it was stated by the authors that this was designed to reduce repetition. That is why their desired situation is $\leq 1$, which is different from [1]. After reading their codes, we realized that [2]'s function could be improved to step-wise penalization. We will clarify these details in the revision.
> > >
> > > 3) [Difference from ours]. The revised function practically-used in [2] still intends to reduce repeats whilst our motivation is to reduce locality bias by aligning attention distributions of the hypothesis and the reference. That is to say, $\sum^t_{m=1} \alpha_{m,i} < g_i, t < T$ is expected to be transformed to $\sum^T_{m=1} \alpha_{m,i} = g_i$ exactly at the end, where the latter is our goal while the former is for better reaching the goal (the desired situation of our method in Table 4 of appendix  may not be very clear, we will modify it). To reach the goal, the global score should work with our length reward (line 149--153). Apparently, our final goal ($=g_i$) is totally different from [2] ($\leq 1$) or [1] ($\geq 1$). Furthermore, we argued in the paper that our motivation follows the essence of generation. That is, if we want to generate a gold summary,  the transformation process of local attention distribution must be in line with our desired situation. If  $g_i$ is replaced with $1$, it would not follow such essence.  As a matter of fact,  we tried [1] or [2] as the baseline, but they performed very poorly in decoding the pre-training model. Actually, the global attention of the pre-training seq2seq model is not relevant with the uniform distribution at all in both summarization and translation. Normally in summarization, only the global attentions of the start token and full stops could surpass $1$, while those of other source tokens are much lower than $1$. We will make more analysis on the global attention distribution in the revision.
> > >
> > >
> > > In summary, we appreciate your comments very much. Hopefully our further response could clarify your concerns.  We would follow your suggestions to improve the paper further. Additional remarks will also be set out in our appendix.

---

> > > > ### Comment · Reviewer_9iis · 2021-09-01
> > > > **response taken**
> > > >
> > > > Regarding attention coverage penalty, while I believe that I've convinced you [2] is using stepwise penalty in their code, after checking tensorflow's code again I found [1] seems to be not using stepwise penalty (https://github.com/tensorflow/addons/blob/96136187b14d6508d42bacbf4c619e789ae04078/tensorflow_addons/seq2seq/beam_search_decoder.py#L1256, I am not absolutely certain since I'm not sure if this is the code for GNMT). Therefore, my concern that the coverage penalty implementation might be wrong for the baselines was not valid (that being said, I think you should change Table 4 in the appendix to reflect the fact that [2] is using stepwise penalty).
> > > >
> > > > Besides, I buy your point that this work is different from bottom-up summarization if $g_i$ (the total attention received by a source token) seldomly exceeds 1 during beam search. It would be nice if you can add some example visualizations of the learned $g_i$'s.
> > > >
> > > > Therefore, I'm increasing my score to 6.

---

### Official Review · Reviewer_ke1n · 2021-07-18

**Rating:** 6
**Confidence:** 3

**Summary:**

This paper presents a method to improve beam search by learning and predicting the global attention. A global scoring function is developed to regulate beam search to generate summaries. In Phase I, the global attention distribution is predicted in order to be included as a protocol to calibrate beam search. In Phase II, a step-wise global scoring function is used to guide beam search based on the predicted global attention distribution.
Experiments on 9 datasets show that the global-aware inference improves state-of-the-art summarization models.


**Limitations And Societal Impact:**

Presentation
Although I believe the idea of this idea is actually pretty straightforward, the presentation and illustration of the idea are hard to follow. There are just so many notations, superscripts and subscripts, like alpha {l,k,t,i}, zeta, G, etc. I believe the presentation can be significantly improved. I would like to see a figure with a toy running example, which I think will walk through how the method works, during training and during inference. Something like source = A B C, decoding prefix = y0 y1, the local attention distribution is [0.1, 0.2, 0.7], the global attention is [....],try to minimize the distance of __ and __, during inference, step 0, step 1, etc.
If the authors want to convince more people to use this method, this step is necessary, from my point of view.
The naming and symbols could be simplified like “local cross attention as the dynamic local attention distribution”.

The method is a little hacky and complicated, and I would like to see more analysis about how attention changes, what’s the global and local attention. The evaluation mostly focuses on ROUGE, but I still don’t have much idea about what happened and what’s the change of the attention pattern or behavior. If this piece can be shown, I believe the paper will seem less hacky because you explain how it works.


**Main Review:**

The angle and direction is pretty interesting. While most of prior work on improving beam search focuses on repetition, similarity/dissimilarity, and length control, this paper presents a relatively new angle. The method proposed can implicitly improve the content planning procedure of text summarization and generation.

The performance improvement is significant. Comparing with other methods (Table 4), it is also competitive.

Question:
Line 96 - 98: “With both sources and  reference targets available in the training set, the attention given by reference tokens is computable, called global cross attention αg ”





**Time Spent Reviewing:**

4

---

> ### Author Response · Authors · 2021-08-10
> **Response to Reviewer ke1n**
>
> Thank you for your suggestions/comments on our paper. In fact, we have been trying our best to polish and improve further our paper, including notations and expressions, even after the paper is submitted. We strongly agree with your suggestion that we need to add a toy running example. We plan to place it in the appendix.
>
> As to the suggestion that this paper need an additional evaluation for attention distribution, we believe that this may not be necessary in our humble opinion. Since ROUGE measures the overlap between reference and hypothesis, then higher ROUGE indicates the hypothesis is more similar to the reference, also means that the attention distribution of the hypothesis is closer to the global attention distribution. Nonetheless, in order to  show more intuitively the change of attention, we will add some case studies on how the global score calibrates beam search stepwise.

---

### Official Review · Reviewer_QnKK · 2021-07-26

**Rating:** 7
**Confidence:** 3

**Summary:**

This paper tackles the problem of suboptimality of beam search as it is not able to have a global perspective of the probability of the generated beams due to truncated beam size (less than vocabulary size) for space and time complexity purposes. This paper proposes a global-aware beam search which is calibrated using the global attention distribution to guide the local decisions at every time step during beam search. The global attention distribution prediction is formulated as a regression task, which is then fed into a global scoring function to guide the beam search-based generation. This approach solves the mismatch between the local optimality of beam search with the global optimal generation. This modified search strategy has been shown to boost the summarization of two SOTA models BART and PEGASUS.


**Limitations And Societal Impact:**

Yes

**Main Review:**

## Strong points:
1. The paper is well-written and clear.
2. The problem is well-motivated and theoretically sound.
3. The idea of using attention distribution to guide beam search in a principled manner is novel and interesting.
4. The experimental result improvements are convincing across all 9 summarization datasets.
5. Thorough analysis has been done to compare with beam search across 9 summarization datasets, in addition to comparison with other methods like coverage, repetition penalty, attention masking, along with ablation studies across various dimensions.
6. The proposed algorithm has also been shown to be robust by using attention prediction from a different dataset.


## Questions:
1. It is not clear (or maybe there is some gap in my understanding), whether the predicted attention distribution still leads to the same theoretical result (Theorem 3.1 and Corollary 3.1.1)
2. It will be interesting to see how this decoding strategy fares in other NLG text2text tasks like Paraphrasing, Neural Machine Translation, etc.
3. How does the proposed strategy compare with other decoding strategies like Top-k Sampling[1], Nucleus Sampling[2]?
4. For completeness, human evaluation with respect to correctness, adequacy, fluency, and coherence will be a helpful comparison.
5. The related work can include more guided decoding strategies as summarized in the blog[3].

## Originality: 7/10.

## Quality: 8/10.

## Clarity: 8/10.

## Significance: 7/10.

REFERENCES:
[1] Hierarchical Neural Story Generation, Fan et al.
[2] The Curious Case of Neural Text Degeneration, Holtzman et al.
[3] Controllable Neural Text Generation: https://lilianweng.github.io/lil-log/2021/01/02/controllable-neural-text-generation.html



**Time Spent Reviewing:**

5

---

> ### Author Response · Authors · 2021-08-10
> **Response to Reviewer QnKK**
>
> Thanks for your valuable comments.
>
>
> ### [Q1: about whether the predicted attention distribution still leads to the same theoretical result]
> Thank you for pointing out this. Through explaining that the global attention value and the predicted attention value have the same value range, we will simply prove that the predicted attention distribution is also applicable to the theory in the paper.
>
> Since each attention value in the global attention distribution is a cumulative value of attention probability $\in [0, 1]$ given by the reference tokens, the lower bound tends to be 0 and the upper bound is 1 * the number of reference tokens.  When the reference is infinite, the upper bound is $\infty$. Therefore, we only need to ensure that the predicted attention value is larger than 0 which is constrained by $\exp$.
>
>
> ### [Q2: about the results of NMT]
> Though the focus of this paper is Neural Abstractive Summarization, we will share how to enable global-aware inference to improve beam search significantly in NMT using a simple variation.
>
> - [a. the gap between NMT and NAS] We don't think the main difference is because there are token overlaps in summarization tasks. Actually, attention could be used in NMT to align the tokens in the source and translation, and these aligned tokens are often semantically "overlapped". We believe that the actual gap is that NAS is a more-to-less generation task where the global attention values of many words are very low, which makes it relatively easy to trigger punishment at the beginning. As you can see in Appendix D.4 and the examples provided, many global-aware summaries are different from that of beam search from scratch, implying the global scoring function is activated in the very early stage. Since translation is a nearly one-to-one task, it is more likely to fail to trigger penalty during the beginning stage when the local attention is generally low.
>
> - [b. how to alleviate this gap] A straightforward-but-effective approach is to transform the one-to-one generation task to several more-to-less tasks. We divide the reference into several blocks of equal length and predict the global attention distribution of each block. During decoding, the local attention in one block is desired not to surpass the global attention of that block. Although this blocking operation would increase the difficulty of training and inference, these negative effects are controllable by adjusting the block length.
>
> - [c. experimental results] We use mBART [1] as the neural translation model whose parameters for WMT16 en-ro sentence-level translation are available at HuggingFace. Compared with beam search, global-aware in this article elevates BELU score from 37.6 to 37.8, and ORACLE global-aware improves the result to 39.2. The blocking operation further improves global-aware to 38.1, and ORACLE global-aware to 41.3. We also experiment this operation on summarization tasks where minor improvement is observed.
>
> This variant was originally part of our follow-up work. Now we decide to put it in the appendix of this article to share the attempt with a larger community of readers. We plan to focus our future study on generalizing the global-aware inference to a broader range of text generation tasks.
>
> ### [Q3: Nucleus Sampling is not applicable to summarization model]
> Thank you for your suggestion. We mentioned Nucleus Sampling in line 308, and argued that it is not applicable to summarization model. In this paper, we discuss the problem of (deterministic) beam search where each step several top successors are selected, and global-aware is designed to improve this kind beam search. Usually, Nucleus Sampling is used for generating text that does not have an optimal hypothesis such as story generation, while in summarization or translation, there is a specific reference for each sample.
>
> ### [Q4 and Q5]
> Thank you very much for your suggestions on supplementing human assessment and related work. We will update these results in the final version.
>
>
> [1] Liu et al., Multilingual Denoising Pre-training for Neural Machine Translation

---

### Official Review · Reviewer_eHvy · 2021-08-02

**Rating:** 7
**Confidence:** 4

**Summary:**

This work proposes to add a global attention feature component to the beam search scoring procedure for conditional generation, which typically just considers the fitness of successor tokens conditioned on the predicted history of tokens in the beam. This augmentation basically keeps track of cumulative local autoregressive attention on the source tokens during beam search and penalizes major deviation from the predicted desired "global" attention over the source tokens. Specifically, this approach proposes a finetuning step of neural language generators, which is responsible for predicting cumulative global attention on each token and the training is done via regression. During training, the "reference" global attention is computed by using the full reference in the decoder transformer instead of just the left prefix of the reference to compute attention.

A relatively simple scheme is proposed to measure deviation of cumulative local attention from the global attention in a decomposable iterative manner. This scheme also involves a form of length reward that encourages the generation to match the length predicted by the global attention.

Experiments are performed on the task fo summarization with pretrained summarization system on 9 datasets. The results show that the proposed beam search augmentation outperforms standard beam search in general. Most strikingly, the ablation experiment with the "Oracle" global attention (instead of prediction, if we had access to the actual global attention and hence the actual reference length) significantly outperforms beam search and the proposed approach demonstrating the potential of global attention.

**Limitations And Societal Impact:**

Yes, I found the sparse discussion sufficient given the mostly algorithmic contribution of this work.

**Main Review:**

Positives:

-- The proposed approach is well motivated, very simple to implement and novel to my knowledge.

-- The results on abstractive summarization show that the proposed approach is effective. Specifically, the oracle experiments clearly show that global attention, although not used during training of the seq2seq models, is a useful signal to model.

Negatives and Questions:

-- The experiments are only on the task of summarization where the token overlap is huge. I would have liked to see the performance of this approach on other related tasks like machine translation.

-- It took me several readings to figure out what exactly was trained/finetuned--and I am not completely sure even now. I believe that the seq2seq models were pretrained and used local attention as is standard during training. The architectures were finetuned to predict global attention. While it is clear that finetuned models were used for global attention prediction, were they also used for decoding or were the pretrained models used for performing the beam search.

-- It is surprising that global attention would be useful given that it is not at all used during training of seq2seq models. Is it mainly useful because of token overlap between source and reference--i.e. while training global attention can use reference to tell the decoder which source tokens are important for the future steps? Again, this is a question that would be better answered by experiments on another task like MT.

-- On the second point, while the proposed approach is compared to beam search, I think there are two beam search baselines to compare to. 1) Beam search with the pretrained BART/PEGASUS, 2) Beam search with finetuned BART/PEGASUS.

-- Also, what happens if the seq2seq training includes global attention (or the full seq2seq training is finetuned with global attention)? This would mean that in addition to finetuning with the regression loss, you would perform finetuning with the actual seq2seq cross entropy loss with global attention.

**Time Spent Reviewing:**

7

---

> ### Author Response · Authors · 2021-08-10
> **Response to Reviewer eHvy**
>
> Thanks for your valuable comments.
>
> ### [Q1: about the results of NMT]
> Though the focus of this paper is Neural Abstractive Summarization, we will share how to enable global-aware inference to improve beam search significantly in NMT using a simple variation.
>
> - [a. the gap between NMT and NAS] We don't think the main difference is because there are token overlaps in summarization tasks. Actually, attention could be used in NMT to align the tokens in the source and translation, and these aligned tokens are often semantically "overlapped". We believe that the actual gap is that NAS is a more-to-less generation task where the global attention values of many words are very low, which makes it relatively easy to trigger punishment at the beginning. As you can see in Appendix D.4 and the examples provided, many global-aware summaries are different from that of beam search from scratch, implying that the global scoring function is activated in the very early stage. Since translation is a nearly one-to-one task, it is more likely to fail to trigger penalty during the beginning stage when the local attention is generally low.
> - [b. how to alleviate this gap] A straightforward-but-effective approach is to transform the one-to-one generation task to several more-to-less tasks. We divide the reference into several blocks of equal length and predict the global attention distribution of each block. During decoding, the local attention in one block is desired not to surpass the global attention of that block. Although this blocking operation would increase the difficulty of training and inference, these negative effects are controllable by adjusting the block length.
> - [c. experimental results] We use mBART [1] as the neural translation model whose parameters for WMT16 en-ro sentence-level translation are available at HuggingFace. Compared with beam search, our proposed global-aware  elevates BELU score from 37.6 to 37.8, and ORACLE global-aware improves the result to 39.2. The blocking operation further improves global-aware to 38.1, and ORACLE global-aware to 41.3. We also experiment this operation on summarization tasks where minor improvement is observed.
>
> This variant was originally part of our follow-up work. Now we decide to put it in the appendix of this article to share the attempt with a larger community of readers. We plan to focus our future study on generalizing the global-aware inference to a broader range of text generation tasks.
>
> ### [Q2: all used seq2seq models are fine-tuned]
> The "fine-tuned" in Line 86 exactly means that the pre-training model has been fine-tuned on a specific dataset. Actually, these fine-tuned parameters of BART and PEGASUS are downloaded from HuggingFace (https://huggingface.co/) and are fixed in **ALL** subsequent operations, including training the attention-prediction model, performing decoding with beam search and global-aware inference.
>
> ### [Q3: about whether the global attention is predictable in other generation tasks]
> In our opinion, we don't think the token overlapping or token importance could play a key role in the attention prediction. An interesting phenomenon is that no matter in BART, mBART or PEGASUS, more attention weights are assigned to special tokens which are not semantically salient. Essentially, global attention contains the model knowledge because it is calculated by the trained model parameters, and these knowledge is regular and traceable. These knowledge tells the decoder how to allocate attention, not only from the perspective of token importance, but also from the features that attention model can capture, such as syntax or word order. Although the model knowledge will be reflected differently in different tasks or data sets, it does not affect what we predict  from source because similar source should have similar attention distribution in most cases, just as they have similar references. In a word, attention distribution is a continuous form of the reference obtained by a series of regular transformations, allowing the reference to be predicted as a whole rather than word-by-word.
>
> ### [Q4]
> Sorry for the confusion. We believe the response to Q2 could solve your question.
>
> ### [Q5: Seq2seq training with global attention]
> We did consider this idea, that is, let the seq2seq model learn the global attention iteratively. However, we temporarily abandoned it because of the efficiency and feasibility issue. It is well-known that there is an exposure bias [2] with teacher forcing training, but we can't deny that it is a very efficient training method. The high efficiency and ease of use make it possible to train expensive large models. Therefore, the main intention of designing this algorithm is to improve inference quality while preserving these hard-won pre-training parameters. As you can see, this scheme is plug-and-play without any model parameter modifications.
>
> [1] Liu et al., Multilingual Denoising Pre-training for Neural Machine Translation
>
> [2] Ranzato et al., Sequence Level Training With Recurrent Neural Networks

---

> > ### Comment · Reviewer_eHvy · 2021-08-25
> > **Thanks for the author response**
> >
> > Thanks a lot for your response to my review. As mentioned in my review, I would like to see some discussion and details about the folowing in the revision:
> >
> > -- It is intriguing that global attention (oracle experiments) seems to help decoding significantly while it is not used during training at all. Infact during training, the "global attention" is never computed-- the closes is the last token prediction when the attention is computed over the input and most of the reference. Also, there is no other signal aside from translation signal to train the attention models during seq2seq training. Then why would it help with decoding. Is global attention interpretable? Does it indirectly indicate the whole target sequence to be decoded? More analysis in this direction would enhance the paper.
> >
> > -- Please explain your training setup in greater detail. You mentioned that you freeze the parameters for attention prediction and decoding. This is an important detail and lack of information in the main draft left me confused.
> >
> > -- Finally, I agree with R9iis that this apporach is not "globally aware" as is conventionally understood. It doesn't explicitly condition its local decoding decisions on potential global decoding decisions. The use of this terminology is confusing and misleading. A simple fix like "awareness of global attention"/ "decoding with global attention" would make the title and terminology more precise and the contributions clearer

---

> > > ### Author Response · Authors · 2021-08-26
> > > **Thanks for your suggestions.**
> > >
> > > -- 1) why would it help with decoding. 2) Is global attention interpretable? 3) Does it indirectly indicate the whole target sequence to be decoded?".
> > >
> > > Thanks for your valuable comments which will further enhance our paper. We will follow your kind suggestions to add more discussions as well as providing more details in our revision. Regarding your questions, we will share our thoughts as follows.
> > >
> > > 1) Since attention model is trained based on the reference tokens in seq2seq training, it is reasonable and reliable to use reference to calculate attention, i.e., the global attention. However, the attention has an exposure bias when it is calculated by generated tokens during inference. This may lead to the locality bias directly. Our work can alleviate such locality bias by aligning the attention distribution between hypothesis and reference. Besides, the oracle results indicated that the global attention is helpful for decoding. The concern left is whether it can be predicted by source. We will conduct more discussions and analysis in the revision. For example, we will plot and discuss the decline trend of training loss and validation loss of our attention training to gain more insight.
> > >
> > > 2) It is interesting to explore if the global attention can be interpretable. To this end, we will visualize in the revision the global attention distribution and examine more cases. As we observed, the most attention paid by a generated token is usually assigned to the start token, and the second is the (semantically) overlapped source tokens.
> > >
> > > 3) We believe the global attention could somehow indicate the target sequence to be decoded. In the revision, we will show how the global attention distribution changes when we adjust some reference tokens, word order, and actual knowledge. In fact, like hidden representations, attention distribution could also capture the "structural similarity" of references, e.g., replacing semantically similar words could bring slighter changes. This is because global attention distribution is exactly a transformation from these hidden states. One advantage is that each feature in such reference "representation" could correspond to a source token, thus we could use the source to predict it conveniently.
> > >
> > > -- We will surely elaborate our training setup in the revision. Particularly, we will change the original expression in line 86 to "Notably, the fine-tuned parameters of both models are downloaded from HuggingFace Models (https://huggingface.co/models) and are fixed in all subsequent operations, where "fine-tuned" means the pre-training model has been fine-tuned on a specific dataset." Other details will also be provided.
> > >
> > > -- Thanks for your suggestion. We will consider to revise the title accordingly.  For example, we will consider to revise the title as "Global attention-aware Beam Search for Neural Abstractive Summarization" or "Decoding with Global Attention for Neural Abstractive Summarization".

---

### Decision · Program_Chairs · 2021-09-27

**Decision:**

Accept (Poster)

**Comment:**

The paper seeks to improve beam search for neural abstractive summarization. It deals with the local optimality problem of the original beam search: at each local step, the assumption is that the optimal hypothesis is within a top-k kept by the beam, but this is often not true. To alleviate this problem, the authors present a global attention mechanism that keeps track of cumulative local attention on the source tokens during decoding while penalizing deviation from the predicted global attention. Experiments on 9 summarization datasets show consistent gains.

The reviewers found many strong points, including a good motivation, a model that is principled and quite novel, and strong empirical results. The main negative points are that the work was not applied to other generation tasks, but extending the approach to, e.g., MT would require several changes to the model (as explained in the author response) and new experiments that would probably not fit in a single paper. Some of the reviewers found some aspects of the presentation confusing, but this could be fixed in the camera-ready version (I agree with the reviewer who suggested adding running example). Related work is a bit short and could mention attempts to mitigate exposure bias (which is not a completely unrelated problem, and its relatedness came up in the discussion) and other ways to get around limitations caused by token-level objectives.

Minor note: The paper makes it sound like beam search was invented in 2016, but it is fundamental algorithm of AI that goes back to much earlier than that. If they want to put a citation, the authors might be better off citing an AI textbook such as Russell & Norvig (or find its earliest use there). If the authors wanted to refer to the first use of beam search in a neural seq2seq setting, I think [Graves, 2012] would be a more appropriate citation (it is used with RNNs, but the core algorithm remains the same). Prior to seq2seq, beam search was used extensively in NLG, including machine translation [Koehn, 2004].

Koehn, 2004: http://homepages.inf.ed.ac.uk/pkoehn/publications/pharaoh-amta2004.pdf

Graves, 2012: https://arxiv.org/abs/1211.3711